

**First Measurement of Atmospheric Mercury Species in Qomolangma Nature**
**Preserve, Tibetan Plateau, and Evidence of Transboundary Pollutant Invasion**
**Authors**
Huiming Lin[1], Yindong Tong[2*], Xiufeng Yin[3,4,5], Qianggong Zhang[4,6], Hui Zhang[7], Haoran Zhang[1],
Long Chen[8], Shichang Kang[3,5,6], Wei Zhang[9], James Schauer[10,11], Benjamin de Foy[12], Xiaoge Bu[2],
Xuejun Wang[1**]
**Affiliations**
1. MOE Laboratory of Earth Surface Processes, College of Urban and Environmental Sciences,
Peking University, Beijing, 100871, China;
2. School of Environmental Science and Engineering, Tianjin University, Tianjin, 300072, China;
3. State Key Laboratory of Cryospheric Science, Northwest Institute of Eco-Environment and
Resources, Chinese Academy of Sciences, Lanzhou, 730000, China;
4. Key Laboratory of Tibetan Environment Changes and Land Surface Processes, Institute of
Tibetan Plateau Research, Chinese Academy of Sciences, Beijing, 100101, China;
5. University of Chinese Academy of Sciences, Beijing, 100039, China;
6. CAS Center for Excellence in Tibetan Plateau Earth Sciences, Beijing, 100085, China;
7. State Key Laboratory of Environmental Geochemistry, Institute of Geochemistry, Chinese
Academy of Sciences, Guiyang, 550002, China;
8. School of Geographic Sciences, East China Normal University, Shanghai, 200241, China;
9. School of Environment and Natural Resources, Renmin University of China, Beijing, 100872,
China;
10. Department of Civil and Environmental Engineering, University of Wisconsin-Madison, WI,
53706, USA;
11. Wisconsin State Laboratory of Hygiene, University of Wisconsin-Madison, WI, 53706, USA;
12. Department of Earth and Atmospheric Sciences, Saint Louis University, MO, 63108, USA;
**Correspondence:**
*Yindong Tong, Tianjin University, Tianjin, China, Email at: yindongtong@tju.edu.cn;
**Xuejun Wang, Peking University, Beijing, China, Email at: wangxuejun@pku.edu.cn;



**Abstract**

Located in the world's 'Third Pole' and a remote region connecting the Indian
Ocean plate and the Eurasian plate, Qomolangma National Nature Preserve (QNNP)
is an ideal region to study the long-range transport of atmospheric pollutants. In this
study, gaseous elemental mercury (GEM), gaseous oxidized mercury (GOM) and
particle-bound mercury (PBM) were continuously measured during the Indian
monsoon transition period in QNNP. A slight increase in GEM concentration was
observed from the period preceding the Indian Summer Monsoon ($1.31\pm0.42$ ng m$^{-3}$)
to the Indian Summer Monsoon period ($1.44\pm0.36$ ng m$^{-3}$), while significant decreases
were observed in GOM and PBM concentrations, decreasing from $35.2\pm18.7$ to
$19.1\pm11.0$ pg m$^{-3}$ and from $30.5\pm12.6$ to $24.7\pm19.9$ pg m$^{-3}$, respectively. A unique
daily pattern of GEM concentration in QNNP was observed, with a peak value before
sunrise and a low value at noon. Unexpectedly, GOM concentrations (with a mean
value of $21.3\pm13.5$ pg m$^{-3}$) in this region were considerably higher than the values in
other clean or even polluted regions. A cluster analysis indicated that the air masses
transported to QNNP changed significantly at different stages of the monsoon, and the
major potential Hg sources shifted from north India and west Nepal to east Nepal and
Bangladesh. With large coverage of glacier in QNNP, local glacier winds could
enforce the transboundary transport of pollutants and transport the polluted air masses
to the Tibetan Plateau. It should be noted that the atmospheric Hg concentrations in
QNNP are higher than the reported values in some background regions, which
addresses the need for a more specific identification of Hg sources in QNNP and the
importance of international cooperation for global Hg controls.
**Keywords**
Indian summer monsoon; atmospheric mercury; trans-boundary transport; glacier
winds; Qomolangma National Nature Preserve

**1. Introduction**

Understanding of atmospheric mercury (Hg) concentration in remote regions is
vital to understand the global atmospheric Hg cycling processes (UNEP, 2013; Angot



et al., 2016; Zhang et al., 2016a). Generally, atmospheric Hg can be divided into three
major types: gaseous elemental Hg (GEM), gaseous oxidized Hg (GOM) and
particle-bound Hg (PBM) (Fu et al., 2010). Over 95% of atmospheric Hg exists in the
form of GEM (Ebinghaus et al., 2002; Huang et al., 2014). Due to its stable chemical
properties and long half-life in the atmosphere (approximately 1-2 years), GEM can
be transported over long distances easily (Fang et al., 2009). In contrast, GOM and
PBM can be deposited quickly from the atmosphere, exposing local environments to
significant impacts (Lindberg and Stratton, 1998; Seigneur et al., 2006; Lynam et al.,
2014). To understand the global and regional cycling of atmospheric Hg, different Hg
monitoring networks and sites have been established in recent decades, such as the
Atmospheric Mercury Network (AMNet) (Gay et al., 2013) and Global Mercury
Observation System (GMOS), which contains over 40 ground-based monitoring
stations distributed in the world (Sprovieri et al., 2016). Generally, atmospheric Hg
background concentrations range between 1.5-1.7 and 1.1-1.3 ng m$^{-3}$ in the northern
and southern hemisphere, respectively (Lindberg et al., 2007; Slemr et al., 2015;
Venter et al., 2015). However, the existing studies are still far from sufficient to obtain
a full understanding of long-range Hg transport due to inadequate monitoring data in
the remote and less-populated regions (Fu et al., 2012a; Zhang et al., 2015a).
The trans-boundary and long-range transport of pollutants have attracted
considerable attention (Zhang et al., 2015b; Li et al., 2016; Pokhrel et al., 2016; Yang
et al., 2018) in the northeastern and southeastern regions of the Tibetan Plateau. The
transboundary invasions of atmospheric pollutants in the Tibetan Plateau have been
evidenced in pollutants such as persistent organic pollutants and black carbon (Zhang
et al., 2015b; Li et al., 2016; Pokhrel et al., 2016; Yang et al., 2018). It was reported
that smoke from biomass burning in the Indian subcontinent could pass through the
natural barrier of the Himalaya (Wang et al., 2015; Pokhrel et al., 2016). HCHs, DDTs
and PCBs were all found to have their highest concentrations in the southeast Tibetan
Plateau during the monsoon season (Wang et al., 2018). Similar conditions have also
occurred for black carbon (Li et al., 2016). However, studies of the trans-boundary



transport of Hg on the Tibetan Plateau are still limited. The existing Hg monitoring
data is affected to varying extents by local emission sources (Fu et al., 2012a; Zhang
et al., 2015; Zhang et al., 2016). The atmospheric Hg concentrations in Waliguan,
located at the northeastern edge of the Tibetan Plateau, originated from or passed
through the urban and industrial areas in Western China and Northern India (Fu et al.,
2012a). Located at the southeastern edge of the Tibetan Plateau, the atmospheric Hg
sources for Shangri-La are Southeast Asia, India and mainland China (Zhang et al.,
2015a). Furthermore, studies are still lacking on trans-boundary transport of Hg in the
Qomolangma National Nature Preserve (QNNP), which directly connects the Indian
Subcontinent and Eurasia. The detailed pollutant transport pathways and seasonal or
daily patterns of atmospheric Hg concentrations in this region are still not clear.

99        QNNP, located at the southern edge of the Tibetan Plateau, is considered one of the

world's cleanest regions (Qiu, 2008). With an average altitude of ~4,500 m a.s.l.,
QNNP is a remote region with sparse human population and rare industries (Qiu,
2008; Yao et al., 2012b; Li et al., 2016). However, it is surrounded by two large
potential pollution sources: the populated and developed eastern China region, which
has experienced about 30 years of quick industrial development, and South Asian
developing countries (e.g., India, Nepal, and Bangladesh), which have been
developing at a quick rate in recent years (Streets et al., 2011; Zhang et al., 2015b;
Yang et al., 2018). China and India are reported as the largest coal consumers in the
world (BP Statistical Review of World Energy 2018), and coal combustion is the
largest source of atmospheric Hg emissions, supplying ~86% of Hg emissions from
fuel combustion (Chen et al., 2016). With QNNP located at the air mass transport
pathway of the Indian Summer Monsoon (ISM) (Li et al., 2016), meteorological
conditions in QNNP vary significantly during the monsoon transition period (Wang et
al., 2001). The monthly average precipitation can range from <50 mm in the non-ISM
period to 950 mm in the ISM period (Panthi et al., 2015). In addition to the monsoon,
with a glacial coverage of ~2,710 km$^2$ in QNNP (Nie et al., 2010), glacier winds could
also have direct effects on the local pollutant transport because glacier winds can





pump down polluted air from the upper levels of the stratosphere to the land surface
(Cai et al., 2007). Therefore, the atmosphere in QNNP is vulnerable to the
surrounding pollution sources (Xu et al., 2009; Li et al., 2016).
To the best of our knowledge, the present work is the first study regarding Hg
monitoring and source identification covering both the preceding-Indian Summer
Monsoon (PISM) and ISM periods in the QNNP. This monitoring site is unique
because it is located in the air mass transport pathway from South Asia to the Tibetan
Plateau. We performed comprehensive and continuous measurements of GEM, GOM
and PBM concentrations during the onset and process period of the Indian monsoon.
To identify the detailed sources, we also combined the real-time Hg monitoring data
with a backward trajectory analysis, clustering analysis and potential source
contribution function (PSCF) analysis. The effects of local glacier winds, caused by
large coverages of QNNP glaciers, on the trans-boundary transport of pollutants were
discussed. This combined monitoring and modeling study could help researchers and
government managers to accurately understand the global Hg cycling process and
potential impacts from the rapidly developing countries in South Asia on the
atmospheric Hg concentrations in QNNP.
**2. Materials and methods**
**2.1 Atmospheric Hg monitoring site**
Atmospheric Hg monitoring was conducted at "Atmospheric and Environmental
Comprehensive Observation and Research Station, Chinese Academy of Sciences on
Mt. Qomolangma" (latitude: 28°21'54" N, longitude: 86°56'53" E) in QNNP, at an
altitude of 4,276 m a.s.l. (Figure 1). In QNNP, Mt. Qomolangma spreads from east to
the west along the border between the Indian subcontinent and the Tibetan Plateau
(Figure 1). Due to the high altitude, QNNP is naturally isolated from the populated
regions, and rare local Hg emission sources have been observed (UNEP, 2013). The
most populated region near this monitoring site is Tingri County (with a population
density of 4 persons per km$^2$), located ~40 km to the southwest of the monitoring site.
QNNP is located in the air mass transport pathway of the ISM (Li et al., 2016), and





the meteorological conditions in QNNP have significant variations between the PISM
and ISM periods (Wang et al., 2001). During the transition period, the temperature in
the Tibetan Plateau and South Asia changes from "southern warm - northern cool" to
"northern warm - southern cool" (Wang et al., 2001). This reverse leads to a
significant increase of diabatic heating over South Asia and the southern slope of the
Tibetan Plateau (Ge et al., 2017), which further affects the wind directions and speeds.
Local glacier winds could also affect the transport of air masses in QNNP. Glaciers
cover ~2,710 km$^2$ in QNNP (Nie et al., 2010), and most of the glaciers are located on
the northern slope of the mountain (Figure 1) (Bolch et al., 2012). The glacier wind is
a continuous downslope wind blowing from glacier surfaces down to the foothills of
the mountain throughout the day. Hence, the transport of air masses in this region is a
combination of atmospheric circulation (monsoon) and local weather conditions
(glacier winds). The structure of the boundary layer over QNNP is also significantly
affected by glaciers (Li et al., 2006). The height of the atmospheric boundary layer
changes significantly in one day from ~350 m above the ground level during the night
to ~2000 m during the day.
**2.2 GEM, GOM and PBM monitoring**
To describe the changes of atmospheric Hg concentrations during the PISM and
ISM periods, the real-time continuous measurements of GEM, GOM and PBM
concentrations were carried out using the Tekran 2537B, 1130 and 1135 instruments
(Tekran Inc., Toronto, Canada) from 15 April, 2016 to 14 August, 2016. During the
operation of the Tekran instruments, ambient air was introduced into the instrument
for 60 minutes through an impactor, a KCL-coated annular denuder, and a Quartz
Fiber Filter (QFF). All the Hg species were converted into Hg(0) and then measured
by cold vapor atomic fluorescence spectroscopy (CVAFS). The collected PBM and
GOM were desorbed in succession to Hg(0) at the temperature of 800 ℃ and 500 ℃,
respectively. Hg-free air was used to flush the 1130 and 1135 systems to introduce the
desorbed PBM and GOM into model 2537B for analysis. The sampling inlet was set
at ~1.5 m above the instrument platform (shown in Figure S1). To mitigate the





impacts of low atmospheric pressures on the pump's train, a low air sampling rate of 7
L min$^{-1}$ for the pump model and 0.75 L min$^{-1}$ (at standard pressure and temperature)
for model 2537B was applied (Swartzendruber et al., 2009; Zhang et al., 2015a;
Zhang et al., 2016a). The Tekran 2537B analyzer was calibrated automatically using
the internal Hg permeation source inside the instrument every 23 h, and the internal
source was calibrated before and after the monitoring by an external Hg source using
a syringe. The Tekran ambient Hg analyzer has been described in more details in the
previous publications (Landis et al., 2002; Rutter et al., 2008; de Foy et al., 2016).

**2.3 Meteorological data**

Throughout the sampling period, the meteorological information was recorded
using the Vantage Pro2 weather station (Davis Instruments, USA) with a 5-minute
resolution. The monitored parameters included the temperature (with a precision of
0.1 ℃), relative humidity (with a precision of 1%), wind speed (with a precision of 0.1
m s$^{-1}$), wind direction (with a precision of 1 °), air pressure (with a precision of 0.1
hPa), solar radiation (with a precision of 1 W m$^{-2}$) and UV index (with a precision of
0.1 MEDs). The snow cover data was obtained from the Moderate Resolution
Imaging Spectroradiometer (MODIS) instrument on board the Terra and Aqua
satellites (MOD10A1, Hall et al., 2010) with a daily 0.05 ° resolution.

**2.4 Backward trajectory simulation**

To identify the atmospheric Hg sources, the Hybrid Single-Particle Lagrangian
Integrated Trajectory (HYSPLIT) model was applied to perform a backward trajectory
simulation (Stein et al., 2015; Chai et al., 2016; Chai et al., 2017; Hurst and Davis,
2017). The HYSPLIT model, known as a complete and mature system for modeling
simple air parcel trajectories of complex pollutant dispersion and deposition, was
developed by the US National Oceanic and Atmospheric Administration (NOAA).
Global Data Assimilation System (GDAS) data with 1 °×1 ° latitude and longitude
horizontal spatial resolution and 23 vertical levels at 6-hour intervals was used for the
backward trajectory simulation. All the trajectory arrival heights were set at 1500 m
above ground level. Every backward trajectory was set for 72 hours in 6-hour





intervals, and the air mass transport regions covered China, Nepal, India, Pakistan and
majority of west Asia. Backward trajectories during the whole monitoring period were
calculated, and cluster analysis was carried out to identify the Hg transport pathways.
The cluster statistics summarize the percentage of back trajectories in each cluster,
and the average GEM concentrations are linked with each cluster. The clustering
algorithm utilized in this study is based on Ward's hierarchical method (Ward Jr,
1963), and minimizing angular distances between corresponding coordinates of the
individual trajectories were chosen to calculate the clusters. By averaging similar or
identical pathways from existing air mass pathways to the receiving site, clusters can
help identify the mean transport pathways of air masses and provide the primary
directions of pollutants transported to the receipting site.
The Potential Source Contribution Function (PSCF) model is a hybrid receptor
model using the calculated backward trajectories to estimate the contributions of
different emission sources in upwind regions and has been applied in many previous
studies (Kim et al., 2005; Kaiser et al., 2007; Fu et al., 2012b; Zhang et al., 2013). The
PSCF calculation is made based on counting the trajectory segments that terminate
within each cell to determine the values for the grid cells in the study domain
(Ashbaugh et al., 1985). In this study, the PSCF model was used to identify the
possible sources of atmospheric GEM. The study domain was separated as $i \times j$ cells.
Then, the PSCF value for the $ij^{\text{th}}$ cell is defined as follows:
$$PSCF_{ij} = \frac{M_{ij}}{N_{ij}}$$

where $N_{ij}$ is the total number of endpoints that fall into $ij^{\text{th}}$ cell during the whole
simulation period, and $M_{ij}$ is the number of endpoints for the same cell that
correspond to GEM concentrations higher than a set criterion. In this study, PSCF
values were calculated based on the average GEM concentration during the whole
sampling campaign. The PSCF value stands for the conditional probability that the
GEM concentration at the measurement site is larger than the average GEM
concentration if the parcel passes through the $ij^{\text{th}}$ cell before it reaches the
measurement site.





To account for and reduce the uncertainty due to low values of $N_{ij}$, the PSCF values
were scaled by an arbitrary weighting function $W_{ij}$ (Polissar et al., 1999). While the
total number of the endpoints in a cell ($N_{ij}$) is less than ~three times the average value
of the end points for each cell, the weighting function will decrease the PSCF values.
In this study, $W_{ij}$ was set using the following piecewise function:

$$W_{ij} = \begin{cases} 1.00 & N_{ij} > 3\,N_{ave} \\ 0.70 & 3\,N_{ave} > N_{ij} > 1.5N_{ave} \\ 0.42 & 1.5N_{ave} > N_{ij} > N_{ave} \\ 0.05 & N_{ave} > N_{ij} \end{cases}$$

Combining the MODIS fire spots data, we used the PSCF analysis to validate the
effects of biomass burning regions. MODIS fire spots data (from 1 April 2016 to 31
August 2016) was obtained from the Fire Information for Resource Management
System (FIRMS) operated by the National Aeronautics and Space Administration
(NASA) of the United States (Giglio et al., 2003; Davies et al., 2004).
**3. Results and discussion**
**3.1 Comparisons of atmospheric Hg concentrations between PISM and ISM**
The GEM, GOM and PBM concentrations at the sampling site were 1.42±0.37 ng
m$^{-3}$, 21.3±13.5 pg m$^{-3}$ and 25.5±19.2 pg m$^{-3}$, respectively, during the whole study
period (Figure 2 and Table 1). GEM accounted for over 95% of all the Hg species.
Figure S2 shows a comparison of the GEM, GOM and PBM concentrations during the
PISM and ISM periods. During the PISM period, the average GEM, GOM and PBM
concentrations were 1.31±0.42 ng m$^{-3}$, 35.2±18.7 pg m$^{-3}$, and 30.5±12.6 pg m$^{-3}$,
respectively, while during the ISM period, the average GEM, GOM and PBM
concentrations were 1.44±0.36 ng m$^{-3}$, 19.1±11.0 pg m$^{-3}$, and 24.7±19.9 pg m$^{-3}$,
respectively. We further compared the Hg concentrations at different ISM stages.
Figure S2 shows that GEM concentrations increased significantly with the
development of the ISM, while decreases of GOM and PBM concentrations were
observed during the study period, with a decrease of 39.0% (the average concentration
change from 20.20 pg m$^{-3}$ to 12.33 pg m$^{-3}$) and 49.6% (the average concentration
change from 21.18 pg m$^{-3}$ to 10.68 pg m$^{-3}$), respectively.



Table 2 summarizes GEM, GOM and PBM concentrations and diurnal variations of
GEM measured by the Tekran system globally. Generally, the GEM concentration in
the QNNP was approaching the reported values in the Northern Hemisphere (~1.5-1.7
ng m$^{-3}$) and was higher than those in the Southern Hemisphere (~1.1-1.3 ng m$^{-3}$)
(Lindberg et al., 2007; Slemr et al., 2015; Venter et al., 2015). Among the global Hg
monitoring sites, the EvK2CNR monitoring site on the southern slope of the Tibetan
Plateau, Nepal, is the nearest station (at a straight-line distance of approximately 50
km) from the monitoring site in this study (Gratz et al., 2013). The average GEM
concentration at EvK2CNR (1.2±0.2 ng m$^{-3}$, from Nov. 2011-Apr. 2012) was slightly
lower than that in the QNNP (1.31±0.42 ng m$^{-3}$ during the PISM period and
1.44±0.36 ng m$^{-3}$ during the ISM period). Compared to Hg concentrations observed at
China's background stations and rural regions (e.g., Waliguan Baseline Observatory
(1.98±0.98 ng m$^{-3}$) (Fu et al., 2012a), Ailaoshan Mountain National Natural Reserve
(2.09±0.63 ng m$^{-3}$) (Zhang et al., 2016a), and Shangri-La Baseline Observatory in
Yunnan province (2.55±0.73 ng m$^{-3}$) (Zhang et al., 2015a)), the average GEM
concentration in the QNNP was lower. However, it should be noted that GOM
concentrations (with a value of 21.3±13.5 pg m$^{-3}$) in this region were much higher
than the values in clean regions (usually lower than 10 pg m$^{-3}$) and a known polluted
region (the suburban area of Beijing (10.1±18.8 pg m$^{-3}$) (Zhang et al., 2013) (Table 2).
One possible explanation for the high GOM concentration is the strong subsidence in
QNNP. The subsidence of the free troposphere would bring GOM-enriched air masses
to the surface layer (Faïn et al., 2009), resulting in the observed high surface GOM
levels (Weiss‐Penzias et al., 2009). In QNNP, with the wide distribution of glaciers,
glacier winds could bring the upper air masses to the land surface layer (Song et al.,
2007), which could further strengthen the subsidence movement.
The increases of GEM concentrations during the ISM period could indicate the
impacts of trans-boundary transport, which has been confirmed by the previous
studies (Fu et al., 2012a; Zhang et al., 2016a). The deposition of GEM from the
atmosphere to the land surface is difficult, and GEM has a much longer residence



time than the other Hg species (Fang et al., 2009), making it a good tracer that can
represent the movement of pollutants. At Ailaoshan in Yunnan province (Zhang et al.,
2016a), a higher TGM concentration during the ISM period ($2.22 \pm 0.58$ ng m$^{-3}$) than
the PISM period ($1.99 \pm 0.66$ ng m$^{-3}$) was also observed. The TGM concentration
during the ISM period ($2.00 \pm 0.77$ ng m$^{-3}$) was also higher than that during the PISM
period ($1.83 \pm 0.78$ ng m$^{-3}$) at Waliguan station in the northeastern Tibetan Plateau (Fu
et al., 2012a). In contrast to GEM, the GOM and PBM levels during the ISM period
were lower than the monitored values during the PISM period (Figure S2 and Table 2).
In previous studies, the PBM concentration in the Kathmandu Valley was lower
during the monsoon period (with a value of $120.5 \pm 105.9$ pg m$^{-3}$) than the
pre-monsoon (with a value of $1855.4 \pm 780.8$ pg m$^{-3}$) and post-monsoon periods (with
a value of $237.6 \pm 199.4$ pg m$^{-3}$) (Guo et al., 2017). In India, PBM concentrations
during the monsoon period (with a value of $158 \pm 34$ pg m$^{-3}$) were lower than that in
the non-monsoon season (with a value of $231 \pm 51$ pg m$^{-3}$) (Das et al., 2016). This fact
could be possibly attributed to precipitation increases brought by the monsoon, which
further causes the wet depositions of PBM from atmosphere. During the ISM period,
the precipitation could increase up to 25% in the South Asia and Tibetan Plateau (Ji et
al., 2011).

**3.2 Diurnal variations of atmospheric Hg species in QNNP**

During the PISM period, all the atmospheric Hg species showed clear diurnal
patterns (Figure 3). For GEM, the minimum concentrations usually occurred at ~12
p.m. (0.87 ng m$^{-3}$, UTC +6 time), while maximum values occurred before dawn (1.98
ng m$^{-3}$ at ~5:30 a.m.). From the afternoon, GEM concentration increased consistently
and reached a peak at sunrise (with a value of 1.98 ng m$^{-3}$). Unlike the daily GEM
changes, GOM and PBM concentrations usually reached maximum concentrations
from ~10:00 a.m. to ~4:00 p.m. in the day, and the concentrations remained relative
stable for the rest of the day. During the ISM period, the diurnal variation of
atmospheric Hg species was less significant compared to the values in the PISM
period. At different stages of the ISM period, the diurnal pattern was also different.





The GEM diurnal variation value decreased over time, from 1.03 ng m$^{-3}$ during the
initial ISM period to 0.43 ng m$^{-3}$ during the final ISM period. For GEM
concentrations during the ISM period, the minimum values all occurred at ~2:00 p.m.,
and the maximum values were observed at ~6:00 a.m. After the sunrise, GEM
concentrations decreased continuously to lower values at noon.
Compared with daily GEM changes in previous studies, the diurnal tendency in
QNNP is unique (shown in Table 2). For the sampling sites in other studies, the
highest GEM concentrations were usually observed during the daytime (Fu et al.,
2008; Mukherjee et al., 2009; Nair et al., 2012; Jen et al., 2014; Karthik et al., 2017).
Kellerhals et al. (2003) reported that the majority of monitoring sites in CAMNet have
a common pattern with the maximum concentration around noon and the minimum
concentration before sunrise. Compared to other observation stations and considering
QNNP as a remote region with high altitude, sparse population and rare industries, the
observed result here may indicate a simple mechanism of variation in GEM
concentration without the complex effect of human activities. Previous studies
suggested that the planetary boundary layer (PBL) could have significant effects on
the concentrations of atmospheric pollutants near the ground (Tie et al., 2007; Han et
al., 2009; Quan et al., 2013). With a large glacier coverage (~2,710 km$^2$), the structure
of the boundary layer over QNNP was significantly affected by glacier winds (Li et al.,
2006). The local PBL may be subject to impacts from the glacier-covered
environment and have a significant diurnal variation. Following sunrise, with the
strengthening of the glacier wind, a strong convection current starts to grow in the
troposphere, and the stock of GEM in the near-ground atmosphere is depleted quickly,
leading to the quick decrease in concentrations. In contrast, after sunset, with the
weakening of the glacier wind, the nocturnal stable boundary layer takes a dominate
position controlling the surface layer, and its height is relatively low (Li et al., 2006),
leading to the accumulation of GEM concentrations.
Comparing the diurnal variations between the PISM and ISM period, the
atmospheric Hg concentrations have almost the same pattern of variations, but the





variation during the ISM period is relatively lower, and the variation becomes less
significant in the later stages of the ISM (Figure 3). The GEM concentration usually
peaked at ~5 a.m. - 6 a.m. in both the PISM and ISM periods. While the peak GEM
concentrations were almost at the same level in the whole period, the decreasing
diurnal variations were mainly due to the increasing GEM concentrations in the
afternoon. The increased GEM concentrations in the afternoon may indicate new
GEM sources in the ISM period. One possible source of GEM in the afternoon might
be Hg(0) reemission from the glaciers. Holmes et al. (2010) reported that
snow-covered land could be a reservoir for the conversion of oxidized Hg to GEM
under the sunlight, and approximately 60% of the Hg deposited to snow cover would
eventually be reemitted to the air. A shorter reservoir lifetime for deposited Hg in
snowpack was also reported when temperature rises (Faïn et al., 2007). With the
increases of ambient temperature and radiation from April to August, the reemission
of GEM from the glaciers could increase as well. As the snow coverage in the QNNP
decreased significantly from the PISM to ISM period (Figure S3), some of the
released Hg may become new GEM sources from the initial ISM to the final stage of
the ISM period. More GEM was released due to the higher temperature and stronger
radiation in afternoon.
**3.3 Source identification for atmospheric Hg in the QNNP**
**3.3.1 Wind direction dependence of Hg concentrations**
Figure 4 shows the concentration roses of GEM, GOM and PBM at the sampling
site during the PISM and ISM period, respectively. All concentrations of the three
species have a strong dependence on the wind directions. During the PISM period, the
predominant wind directions with Hg masses are northeast and southwest. Wind from
the northeast of QNNP originates from and/or passes through other parts of China.
The southwest wind, which is the dominant direction and contains the largest amount
of Hg, potentially brought air masses from India and Nepal to QNNP. During the ISM
period, the predominant wind directions with Hg changed to the south and northeast.
Considering the transport rates of species Hg concentrations (length of sector) from



different directions, both directions may have greatly contributed to the Hg
concentration in QNNP, while the air masses from south brought relatively larger
amounts of GOM and PBM.
Relatively low GEM concentrations (<1.5 ng m$^{-3}$) were observed in most of the
samples (80.0%) of air masses in the predominant Hg-transport direction (from
southwest to west) during the PISM period, which is due to the control of westerlies.
With high wind speed (Table 1) and coming from Central Asia, the westerlies are the
predominant wind containing low pollutant levels that spread in the QNNP during the
PISM period (Kotlia et al., 2015). Relatively high GEM concentrations (>1.5 ng m$^{-3}$)
were found in 92.4% of the samples for the predominant Hg direction during the ISM
period under the control of the monsoon (Kotlia et al., 2015), which might indicate
that the transported air masses are coming from polluted regions. GOM and PBM had
similar patterns under the control of the westerlies and monsoon during the PISM and
ISM period, respectively.
**3.3.2 Air mass back trajectories analysis**
To further quantify the contributions of different sources to GEM concentrations,
an air mass back trajectory simulation and trajectory cluster analyses were applied in
this study. Figure 5 provides the trajectory clusters of GEM during the PISM and ISM
periods. According to the total spatial variation index, all the trajectories were
grouped into 6 clusters. During the PISM period (Figure 5a), GEM concentration
from cluster 3 (with the frequency of 17%) was the highest (1.36 ng m$^{-3}$), which
originated from or passed through central Asia, northern India and northwestern Nepal.
Cluster 2 (14%), cluster 5 (19%) and cluster 6 (19%) represent the air masses that
pass through northern India and northwestern Nepal. According to the local Hg
emission inventory (UNEP, 2013), Hg in this air mass most likely originated from
central Pakistan and northern India. Cluster 4 (29%) represents the air masses that
originated from or passed through different cities in northern India. Based on the
previous atmospheric Hg emission inventories (UNEP, 2013; Simone et al., 2016), Hg
emission in the west Asia and central Asia is not significant. Based on a combination



of the pathway analysis, emission inventory and GEM concentration during the PISM
period, almost all the GEM delivered by air masses to QNNP was from northern India
and passed through Nepal.
During the ISM period (Figure 5b-5f), the transport pathways of atmospheric Hg
changed signally with the monsoon onset process of the ISM and differed strongly
from the PISM period. During the ISM1 period (Figure 5b), the onset of the ISM was
under development, leading to the scattered clusters. GEM levels in cluster 3 (21%)
were the highest (1.51 ng m$^{-3}$), which originated from or passed through the Tibetan
Plateau. Cluster 4 (13%), cluster 2 (17%) and cluster 6 (38%) represent the pollutant
coming from Nepal, and the trajectory is relatively short. During the ISM2 period, all
the clusters originated from or passed through central Asia, northern India and
northwestern Nepal (Figure 5c). The clusters were similar to most of the clusters
during the PISM period; however, the GEM concentrations in these clusters were
higher than those during the PISM period. During the ISM3 period (Figure 5d), most
of the clusters moved from west to south of QNNP. Cluster 2 (1.56 ng m$^{-3}$, 44%)
represents the pollutant coming from Bangladesh and passing through southeastern
Nepal. Cluster 3 (1.62 ng m$^{-3}$, 33%) originated from or passed through central Nepal.
The share of air masses coming from central Asia, northern India and northwestern
Nepal dropped to approximately 22%. During the ISM4 period (Figure 5e), the
clusters moved further west to Bangladesh and eastern India. Except for cluster 4
(6%), the other clusters originated from or passed through Bangladesh, eastern India
and northeastern Nepal. The condition during the ISM5 period was almost the same as
the ISM4 period: pollutants were coming from Bangladesh and eastern India and
passed through southeastern Nepal.
PSCF models were also applied to identify the potential sources by combining the
backward trajectory simulation and Hg monitoring concentrations. Figure 6 shows the
regional contributions of GEM emission sources during the PISM period and ISM
period (ISM1-5). During the PISM period (Figure 6a), most of the Hg sources were in
Pakistan, northern India and central Nepal (Zhang et al., 2015a). The QNNP was most





likely impacted by the Hg emissions in Karachi, Lahore (Pakistan), New Delhi, Uttar
Pradesh (India), Katmandu and Pokhara (Nepal), all of which are large urban regions
with intensive industrial activities. With the development of the ISM, the potential
sources gradually shifted from western Nepal to eastern Nepal and Bangladesh
(Figure 6b-f). The PSCF analysis indicated that the air masses could have
transboundary transport events from Pakistan, India, Nepal and Bangladesh to QNNP.
Atmospheric Hg clusters during both the PISM and ISM periods indicated that the
air masses, which originated from or passed through northern India and Nepal, would
make great contributions to the Hg concentration in the QNNP. Northern India and
Nepal were also identified as potential source regions for QNNP. Clusters 2-6 of the
PISM period represent the air masses from outside China, and they show that over 97%
of the GEM in QNNP was transported from outside China during the PISM period.
During ISM2-5 the period, over 95% of the GEM was transported to QNNP from
outside China. Meanwhile, the GEM concentration increased by 10% from the PISM
to ISM period according to the site monitoring data, indicating the increasing amount
of transported GEM. According to the UNEP Hg emission inventory (UNEP, 2013),
northern India is an important Hg source which might be responsible for the
trans-boundary transportation of Hg to China (Figure 5), and the growing emissions in
India are related to the rapidly growing economy and increasing usage of fossil fuels
(Sharma, 2003). Considering the heavy air pollution in Nepal (Forouzanfar et al.,
2015; Rupakheti et al., 2017), Nepal might be an underestimated Hg source in the
modeling and should be taken into consideration in further work.
Under the control of the ISM during the ISM2 period, the high PBM concentration
may be related to the biomass burning in the source region. According to the PSCF
analysis, northern India and Nepal are the potential source regions during the ISM2
period. The source identification by back trajectory simulation and trajectory cluster
analyses also indicated that northern India and Nepal are in the air mass transport
trajectory that would transport Hg to QNNP. Finley et al. (2009) reported that PBM
concentrations may associated with Hg emissions from wildfire events. One possible



cause of the observed high PBM concentration is the frequent fire events that
occurred during the ISM2 period in the air masses trajectory. Figure S3 shows the fire
hotspots observed by MODIS from April to August 2016. During the ISM2 period,
frequent fire hotspots were identified in the source region, and large amounts of PBM
were released into the atmosphere from biomass burning. The transport of those air
masses with enriched PBM was controlled by the ISM and intensified by glacier
winds. The transport of polluted air to QNNP resulted in the outburst of PBM
concentration during the ISM2 period. During the PISM period, although the number
of fire hotspots was much higher, most of the fire hotspots locations were not in the
potential source region (Figure 6a and Figure S4), resulting in the low GOM and
PBM concentrations observed.
**3.4 Implications from this study**
At a high altitude and located in the deep southern Tibetan Plateau, QNNP is
isolated from anthropogenic perturbations and industrial activities, and this area was
thought to be shielded from pollutant inputs from South Asia. However, our results
show that the Hg concentration in this region is not as low as previously expected.
During the whole monitoring period, the highest GEM concentration reached 3.74 ng
$m^{-3}$, ~2.5 times higher than the average concentration in the Northern Hemisphere
(~1.5-1.7 ng $m^{-3}$) (Lindberg et al., 2007; Slemr et al., 2015; Venter et al., 2015). The
average GEM concentration in the middle stage of the ISM was 1.56 ng $m^{-3}$, which is
inside the average range of observed Northern Hemisphere GEM concentrations.
Even considering the PISM period, which is a period of relatively lower GEM levels
in QNNP, the average GEM concentration (1.31±0.42 ng $m^{-3}$) was at the same level
as some monitoring stations in the Northern Hemisphere (e.g., 1.35±0.17 ng $m^{-3}$ in a
rural site in Atlanta, USA).
We now recognize that trans-boundary transportation is an important mechanism
that can influence Hg distribution in this region. In particular, the air masses
transported to QNNP might be primary under the control of mesoscale ISM drivers
and intensified by regional glacier winds (Figure 7). From the PISM to ISM periods,



the warm center gradually shifts northwestward from low latitudes to the QNNP (Wang et al., 2001; Ge et al., 2017), and the South Asian High moves onto the Tibetan Plateau and maintains a strong upper-level divergence and upward motion. The upward motion makes the air masses cross the high-altitude Himalayan mountains and move to mainland China (Xu et al., 2009; Bonasoni et al., 2010). The transboundary transported air masses can be pumped down right after crossing Mt. Qomolangma due to the control of the regionally unique wind transportation mode, the glacier wind. Hence, in addition to the monsoon, the trans-boundary transport of Hg could also be intensified by regional glacier winds, leading to the increases of atmospheric Hg in this region. As in other studies in the northern or eastern Tibetan Plateau, the glacier wind can pump down air masses from stratosphere to the surface in QNNP (Cai et al., 2007). The pump movement is remarkably efficient at transporting air masses (Lelieveld et al., 2018), bringing a considerable amount of pollutants to QNNP.

In 2013, the Minamata Convention on Mercury was developed to control global Hg pollution (Minamata Convention on Mercury). Atmosphere Hg has been reported (Zhang et al., 2016b) to have strongly declined (~1–2% $y^{-1}$) . Under the Convention, a National Implementation Plan on Mercury Control has also been developed in China to fulfill the commitment to control and reduce Hg emissions (World Bank, 2016). GEM concentrations in East China decreased from 2.68±1.07 ng $m^{-3}$ in 2014 to 1.60±0.56 ng $m^{-3}$ in 2016 (Tang et al., 2018). However, the source identity analysis in QNNP indicates that foreign regions of China were the main contributor responsible for the observed pollutants (accounting for 95% of the whole trajectory during the main ISM period). This result indicates that the Hg concentration in QNNP could hardly benefit from China's efforts toward Hg reductions. South Asian developing countries (e.g., India, Nepal, and Bangladesh) (Streets et al., 2011; Zhang et al., 2015b; Yang et al., 2018) should be the key to controlling atmospheric Hg concentrations in QNNP. Hg emissions in India were estimated to be approximately 310 tons in 2010 and are predicted to rise to 540 tons in 2020 (Burger Chakraborty et al., 2013). It is



urgent for those countries to take immediate actions to reduce Hg emissions, which is
crucial to reduce the atmospheric Hg concentrations in QNNP.

**4. Conclusions**

A comprehensive investigation of the concentrations, origin and transport of GEM,
GOM and PBM was made in QNNP, a remote, high-altitude station located at the
boundary between the Indian subcontinent and the Tibetan Plateau and in the transport
pathway of the ISM from South Asia to the Tibetan Plateau. The average GEM
concentration ($1.31 \pm 0.42$ ng m$^{-3}$) during the PISM period was lower than that
($1.44 \pm 0.36$ ng m$^{-3}$) during the ISM period. The average GOM and PBM
concentrations during the PISM period were higher than those during the ISM period,
which might be related to the increasing wet depositions during the ISM period. The
average GOM concentration was higher than in most rural areas in the US and China.
The GEM concentration had a significant diurnal variation pattern in QNNP, with the
maximum GEM concentration observed before sunrise and a sharp decrease after
sunrise until noon. The range of the diurnal variation declined from April to August,
which could be related to the re-emission of Hg from snow cover and melted snow.
According to the backward trajectory analysis and cluster analysis, most of the air
masses with high GEM concentrations in QNNP originated from or passed through
Bangladesh, northern India and central Nepal. With the PSCF analysis, we found that
Pakistan, northern India and Nepal are potential source regions during the PISM
period, and Bangladesh, north India, Nepal were identified as outbound potential
sources during the ISM period. During the ISM period, the air masses would cross the
high-altitude Himalayan mountains with the help of the ISM. Once the air masses
passed through Himalaya, they could be trapped in the surface layer and transported
to QNNP by the all-day-long downslope glacier wind. It should be noted that the
atmospheric Hg values in QNNP were contaminated and even higher than the
reported values in some background regions. Because Hg is easily transported long
distances via the atmosphere, the nations in South Asia must work together to develop
and apply appropriate pollutant-reduction strategies to reduce Hg emission.





**Acknowledgments**

This study was funded by the National Natural Science Foundation of China (Grant #41630748, 41501517, 41571130010) and the Natural Science Foundation of Tianjin (Grant #16JCQNJC08300). The authors are grateful to NOAA for providing the HYSPLIT model and GFS meteorological files, to NASA for providing the MODIS files. We also thank the staffs of the Atmospheric and Environmental Comprehensive Observation and Research Station of Chinese Academy of Sciences on Mt. Qomolangma for field sampling assistance.

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





**Figure captions**

Figure 1. Location of monitoring site in this study (QNNP);

Figure 2. Changes of GEM, GOM and PBM concentrations during the study period;

Figure 3. Diurnal variations of GEM, GOM and PBM concentrations during the PISM and ISM

period;

Figure 4. Concentration roses of GEM, GOM and PBM on different wind directions;

Figure 5. Back trajectories analysis at the monitoring site during the PISM period (a) and the ISM

period (b-f);

Figure 6. Potential source regions and pathways of GEM at monitoring site by the PSCF during

the PISM period (a) and the ISM period (b-f);

Figure 7. Concept maps for trans-boundary transport of atmospheric Hg

**Table captions**

Table 1. The statistics of GEM, GOM, PBM and meteorological variables in different episodes at

QNNP

Table 2. Summary of atmospheric Hg concentration and diurnal variation in previous studies and

this study





**Figure 1**

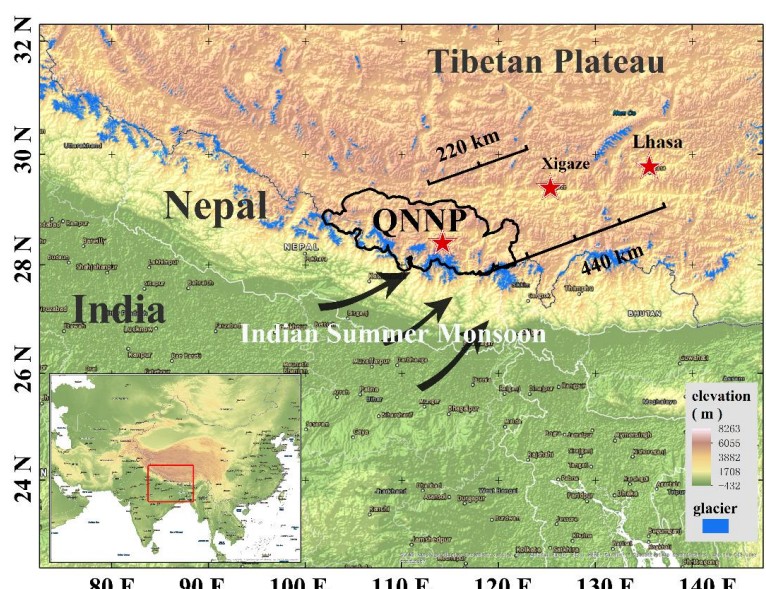






**Figure 2**

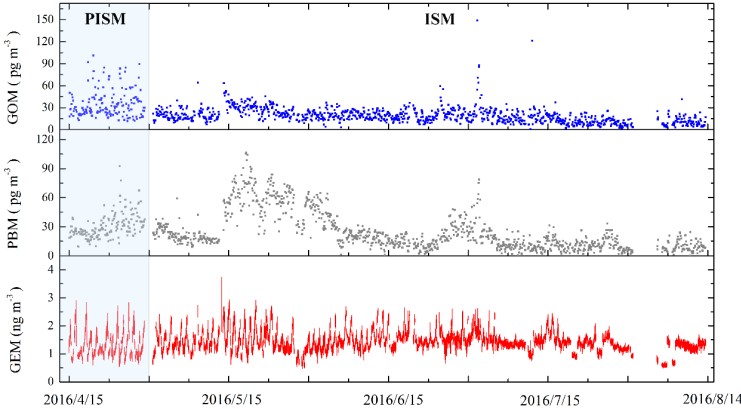







**Figure 3**

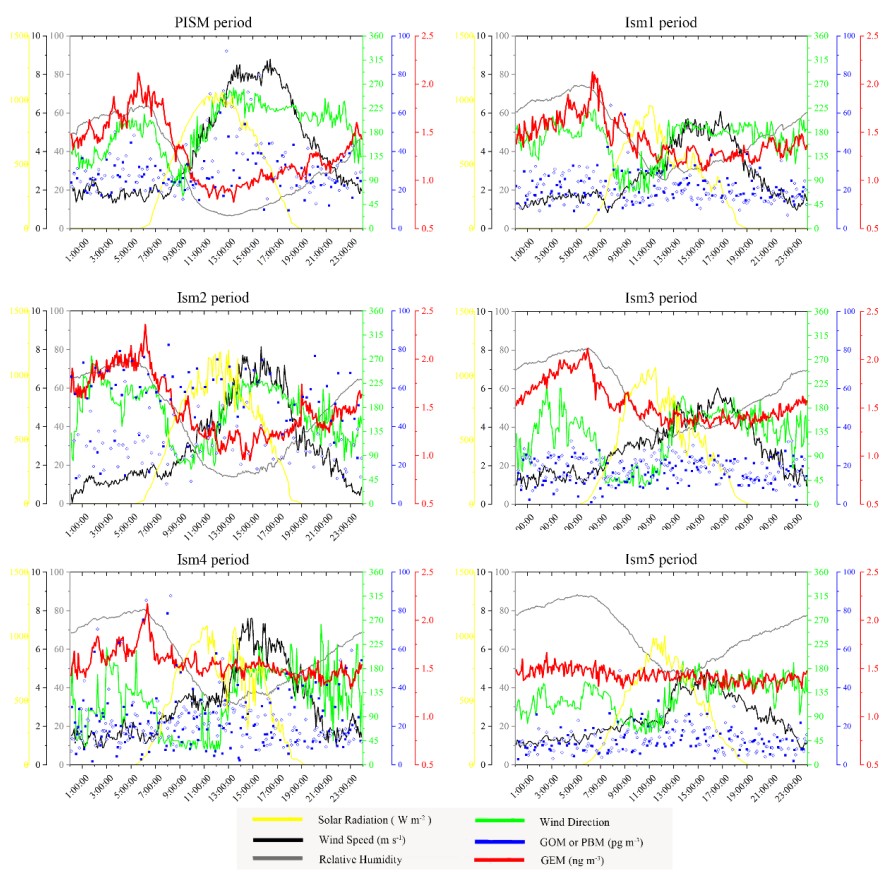






**Figure 4**

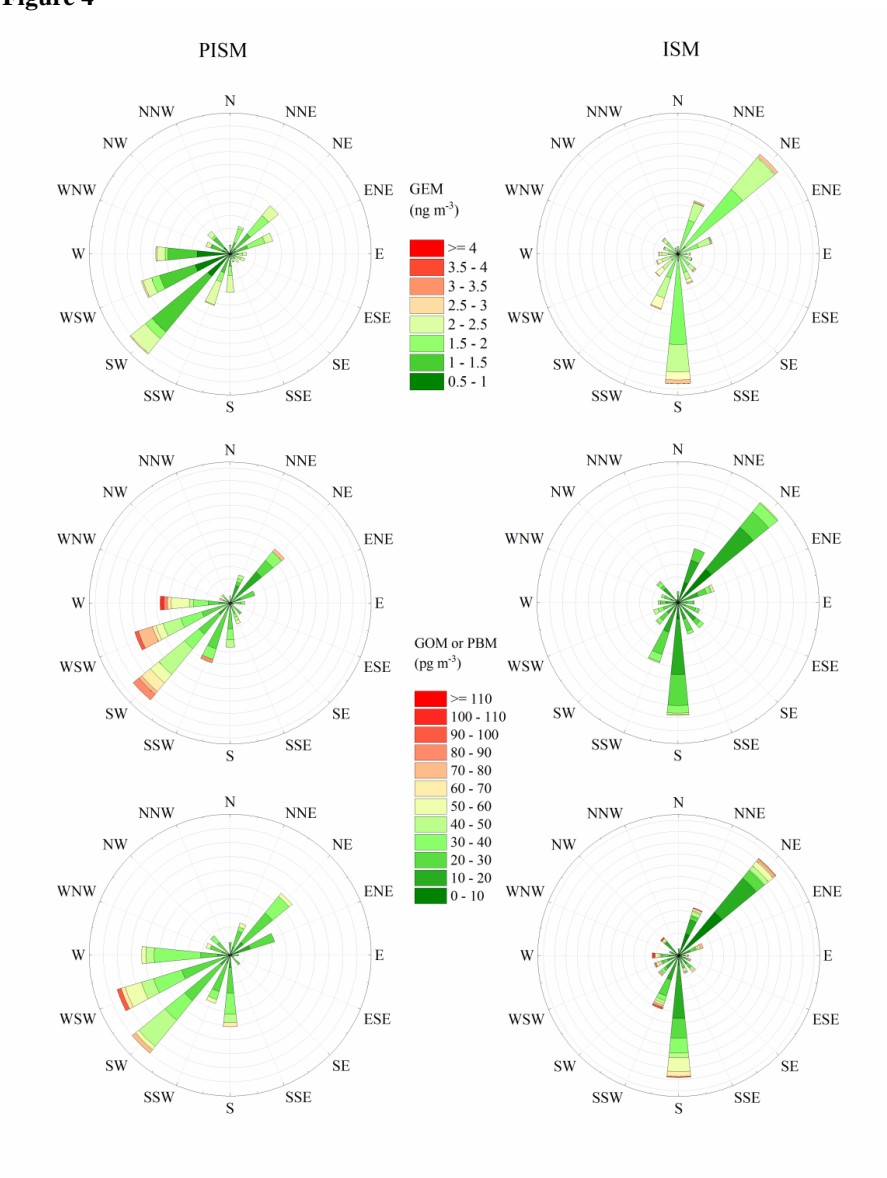







**Figure 5**

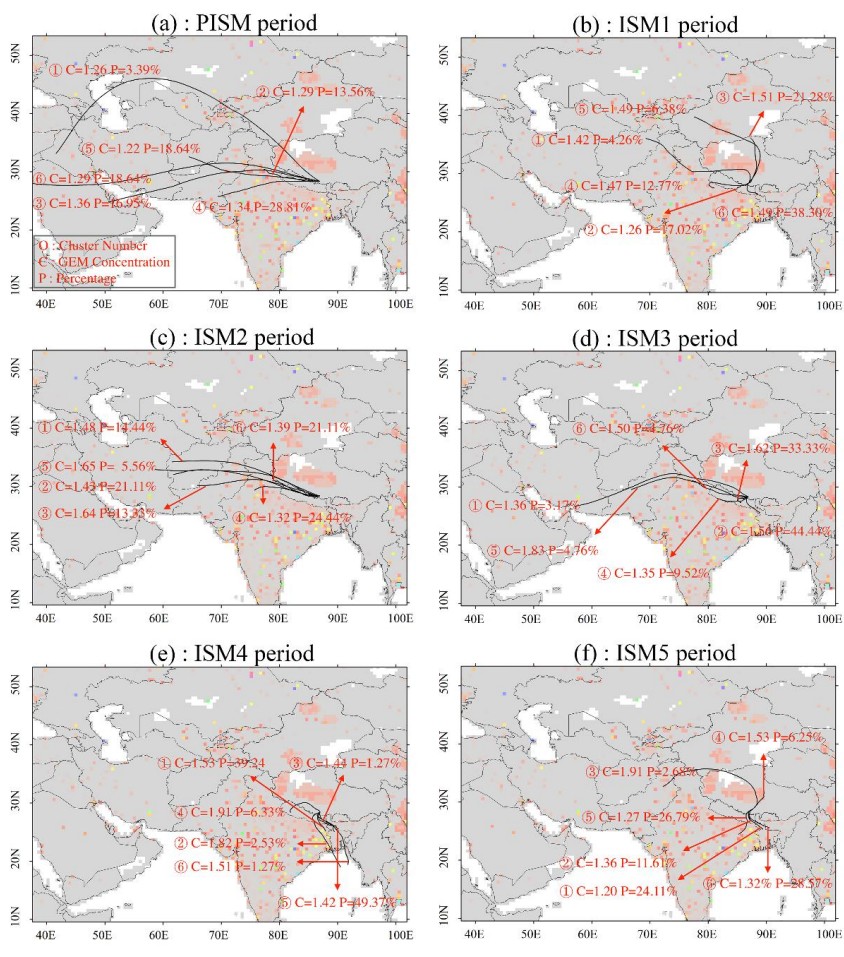





**Figure 6**

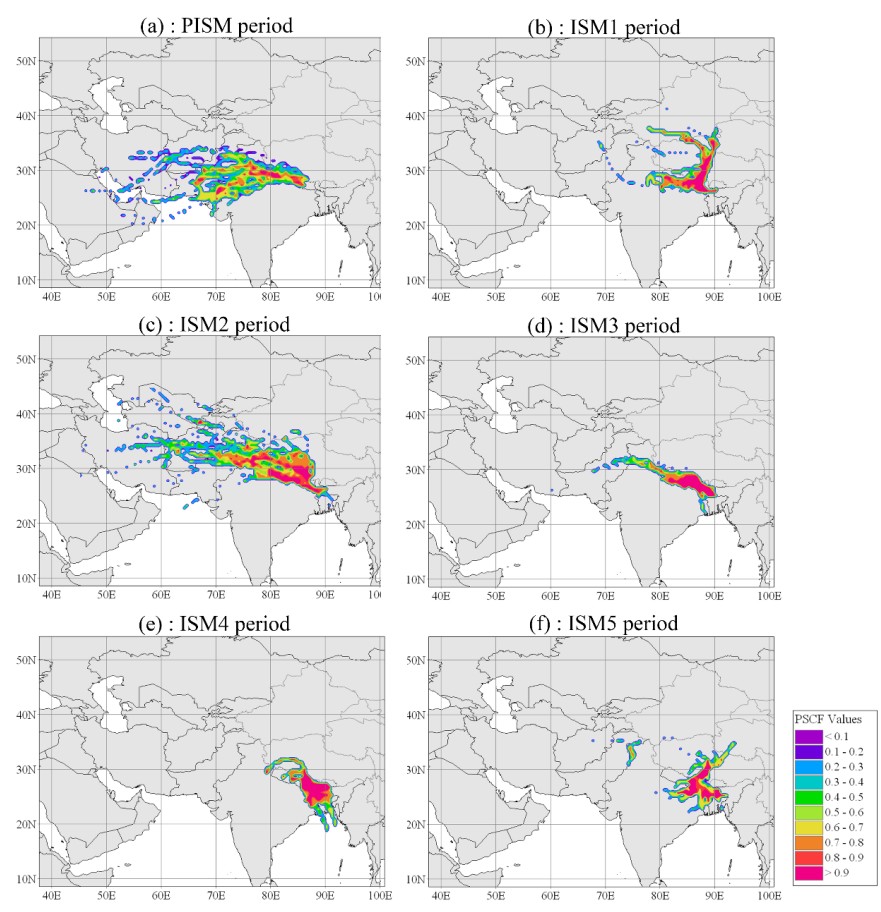









**Figure 7**

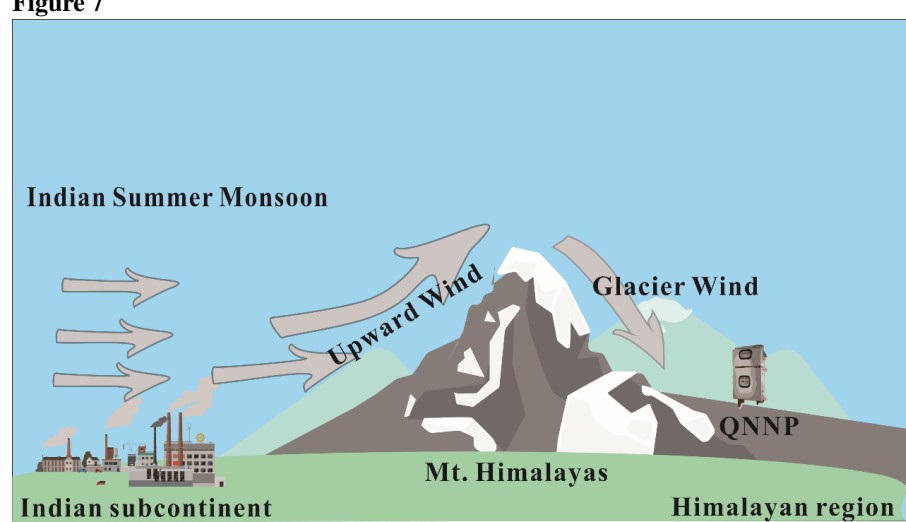







**Table 1. The statistics of GEM, GOM, PBM and meteorological variables in different**

**episodes at QNNP**

| Period | Statistical | T (C°) | RH(%) | WS(m s⁻¹) | GEM (ng m⁻³) | GOM (pg m⁻³) | PBM (pg m⁻³) |
|--------|-------------|--------|-------|-----------|--------------|--------------|--------------|
| PISM | Minimum | -5.6 | 1 | 0 | 0.54 | 11.6 | 9.4 |
|  | 1st Qu. | 1.6 | 11 | 1.8 | 0.99 | 21.7 | 22.3 |
|  | Median | 6.4 | 25 | 3.6 | 1.19 | 29.5 | 26.9 |
|  | Mean | 6.1 | 33 | 4.1 | 1.31 | 35.2 | 30.5 |
|  | 3rd Qu. | 11.2 | 53 | 6.3 | 1.58 | 42.8 | 36.1 |
|  | Maximum | 16.3 | 89 | 13.9 | 2.91 | 101.3 | 92.6 |
| ISM1 | Min | -3.8 | 9 | 0 | 0.15 | 7.1 | 9.1 |
|  | 1st Qu. | 1.6 | 33 | 1.3 | 1.20 | 15.0 | 17.0 |
|  | Median | 5.6 | 49 | 2.2 | 1.38 | 19.2 | 19.1 |
|  | Mean | 5.6 | 50 | 2.7 | 1.44 | 20.2 | 21.2 |
|  | 3rd Qu. | 9.8 | 65 | 3.6 | 1.63 | 24.1 | 24.5 |
|  | Max | 15.7 | 91 | 10.3 | 2.74 | 64.0 | 59.1 |
| ISM2 | Min | -1.3 | 3 | 0 | 0.47 | 3.9 | 12.4 |
|  | 1st Qu. | 4.1 | 30 | 1.3 | 1.14 | 18.5 | 40.4 |
|  | Median | 8.5 | 48 | 2.2 | 1.35 | 23.7 | 54.8 |
|  | Mean | 8.8 | 46 | 3.0 | 1.45 | 25.4 | 53.4 |
|  | 3rd Qu. | 13.7 | 64 | 4 | 1.68 | 31.3 | 64.9 |
|  | Max | 19.6 | 87 | 11.2 | 3.74 | 63.4 | 106.3 |
| ISM3 | Min | 2.6 | 26 | 0 | 0.78 | 3.2 | 0.8 |
|  | 1st Qu. | 8.1 | 44 | 1.3 | 1.33 | 14.5 | 12.4 |
|  | Median | 11.8 | 58 | 2.7 | 1.51 | 18.9 | 17.1 |
|  | Mean | 12.0 | 58 | 2.9 | 1.56 | 19.2 | 16.8 |
|  | 3rd Qu. | 15.6 | 73 | 4 | 1.72 | 23.4 | 22.0 |
|  | Max | 21.8 | 92 | 9.9 | 2.70 | 36.6 | 31.3 |
| ISM4 | Min | 6.0 | 25 | 0 | 0.66 | 6.7 | 0.3 |
|  | 1st Qu. | 9.3 | 43 | 1.3 | 1.35 | 12.9 | 10.6 |
|  | Median | 12.1 | 61 | 2.7 | 1.46 | 18.0 | 17.3 |
|  | Mean | 13.0 | 58 | 2.9 | 1.51 | 21.0 | 20.0 |
|  | 3rd Qu. | 16.6 | 72 | 3.6 | 1.61 | 24.9 | 26.1 |
|  | Max | 22.7 | 90 | 9.9 | 2.62 | 149.1 | 78.6 |
| ISM5 | Min | 2.2 | 18 | 0 | 0.48 | 0.8 | 0.2 |
|  | 1st Qu. | 8.3 | 59 | 0.9 | 1.17 | 7.2 | 6.2 |
|  | Median | 10.7 | 75 | 2.2 | 1.35 | 10.7 | 9.4 |
|  | Mean | 11.4 | 72 | 2.3 | 1.32 | 12.3 | 10.7 |
|  | 3rd Qu. | 14.1 | 86 | 3.1 | 1.49 | 16.0 | 14.1 |
|  | Max | 22.9 | 96 | 9.4 | 2.45 | 121.3 | 33.2 |





**Table 2. Summary of atmospheric Hg concentration and diurnal variation in previous studies and this study**

| Location | Elevation | Classification | Time period | GEM/(TGM) (ng m⁻³) | GOM (pg m⁻³) | PBM (pg m⁻³) | GEM diurnal variation (Local time/GEM Conc.) | | | reference |
|---|---|---|---|---|---|---|---|---|---|---|
| | | | | | | | peak | valley | variation value | |
| Yorkville, Atlanta, USA | 395 | rural | 2007-2008 | 1.35±0.17 | 8.55±18.8 | 4.43±5.59 | 11/1.37 | 6/1.32 | 0.05 | (Nair et al., 2012) |
| Birmingham, Alabama, USA | - | urban | 2005-2008 | 2.12±1.57 | 78.2±441.9 | 39.5±147.9 | 9/2.27 | 16/1.87 | 0.40 | (Nair et al., 2012) |
| Pensacola, Florida, USA | - | rural | 2005-2008 | 1.35±0.18 | 4.24±6.90 | 2.49±2.87 | 10/1.38 | 5/1.29 | 0.09 | (Nair et al., 2012) |
| Mt. Waliguan, China | 3816 | remote | Sep 2007-Sep 2008 | (1.98±0.98) | 7.4±4.8 | 19.4±18.1 | 6/2.3 | 14/1.94 | 0.36 | (Fu et al., 2012a) |
| Mt. Leigong, China | 2178 | remote | May 2008-May 2009 | 2.80±1.51 | - | - | 14/2.99 | 5/2.52 | 0.47 | (Fu et al., 2010) |
| Mt. Gongga, China | 1640 | remote | May 2005-July 2006 | (3.98) | - | - | 11/4.45 | 2/3.55 | 0.90 | (Fu et al., 2008) |
| Kodaikanal, India | 2343 | rural | Nov 2012-Sep 2013 | (1.53±0.21) | - | - | 16/1.66 | 7/1.43 | 0.23 | (Karthik et al., 2017) |
| EvK2CNR, Nepal | 5050 | remote | Nov 2011-Apr 2012 | (1.2±0.2) | - | - | 18/1.3 | 6/1.1 | 0.1 | (Gratz et al., 2013) |
| Shangri-La, China | 3580 | remote | Nov 2009-Nov 2010 | (2.51±0.73) | 8.22±7.9 | 38.32±31.26 | 17/2.48 | 6/1.71 | 0.77 | (Zhang et al., 2015) |
| Colorado | 3220 | remote | April 2008-July 2008 | 1.6±0.3 | 20±21 | 9±6 | | | | (Fa`ïn et al., 2009) |
| Miyun, China | 220 | rural | Dec 2008-Nov 2009 | 3.22±1.74 | 10.1±18.8 | 98.2±112.7 | 20/3.40 | 10/3.00 | 0.40 | (Zhang et al., 2013) |
| Penghu Islands, China | 25 | coastal | Mar 2011-Jan 2012 | (3.17±1.17) | - | - | 11/3.48 | 1/2.87 | 0.61 | (Jen et al., 2014) |
| Namco, China | 5300 | remote | Nov 2014-Mar 2015 | 0.95±0.19 | - | - | 8/1.00 | 17/0.92 | 0.08 | (de Foy et al., 2016) |
| ALS, China | 2450 | remote | May 2011-May 2012 | (2.09±0.63) | 2.3±2.3 | 31.3±28.4 | - | - | - | (Feng and Fu, 2016) |
| QNNP, China (this study) | 4267 | remote | Apr 2016-Aug 2016 | 1.42±0.37 | 21.3±13.5 | 25.5±19.2 | 6/2.04 | 13/1.11 | 0.93 | This study |