# Peer review of "First Measurement of Atmospheric Mercury Species in Qomolangma Nature"

_Atmospheric Chemistry and Physics, 2018_

## Referee Comment (RC1) · Anonymous Referee #2 · 28 Sep 2018

The authors present speciated Hg measurements (GEM, GOM, and PBM) at a high-altitude station in Tibet near the border to Nepal. They show a pronounced concentration differences between pre-monsoon and monsoon periods and explain them by changing transport patterns encompassing different source regions, especially those in Pakistan, India, and Bangladesh. They also show influence of biomass burning.

There are only a few measurements in this part of the world and, thus, they deserve to be published. Their interpretation is sound. Unfortunately, the data presentation is marred by at times awkward wording, imprecise citation of references, uninformative
figure captions, etc., and thus it needs a good deal of editing. Some improvements are proposed below.

Factual comments

Section 2.2: This section describes essentially the GOM and PBM measurement but not the measurement of GEM. Sampling time for GEM measurements has to be stated. The reason is that the GEM (with usually 5 min sampling), GOM, and PBM data are probably biased low due to problems with the internal default integration because less than 10 pg was collected for the individual analysis (Slemr et al., 2016; Ambrose, 2017). This problem is especially important at the QNNP station because only flow rates of 0.75 and 7 l(STP) min-1 were used for GEM and GOM/PBM measurements, respectively, instead of the usual 1 and 10 l(STP) min-1. The authors should mention the bias and assess its average magnitude using Fig. 3 of Slemr et al. (2016). This is needed when the data are compared to measurements at other sites. A definition of standard pressure and temperature would be also helpful.

Section 2.4: The use of backward trajectories for identification of the source areas seems to me to be questionable in this particular case. If I understand it properly the trajectory arrival height was set 1500 above the station, i.e. at an altitude of some 5800 m. In addition, the station is located in a very complex terrain (mountains above 8000m) with local winds due to glacier coverage. The question is how well the trajectories are representative for the air analysed at the station? Can the authors say anything about it?

Section 3.1: Averages and standard deviations should always be given with the number of measurements since only with it the significance of the differences can be determined. Are the difference of GEM, GOM and PBM concentration between PISM and ISM periods statistically significant?

Lines 278-283: Subsidence is probably only a part of the explanation, lack of precipitation could be another part.

Table 2 claims to summarise global measurements of GEM, GOM, and PBM which is far from being true. Outside of Asia only three US sites are listed which is only a small fraction of all measurements (Sprovieri et al., 2010, 2017; Gay et al., 2013). In addition, these three US sites are not mentioned in the text. Since a comprehensive list would fill several pages I would recommend to concentrate on the measurements in Asia and for comparison with worldwide concentrations only to refer to above references.

Section 3.2: In the text a sum of GOM and PBM is discussed but in the legend of Figure 3 symbols are declared as PBM or GOM. Please correct. The caption of Figure 3 reads as if the presented diurnal variations were representative of different periods, i.e. as averages of several days, but the reader has an impression that diurnal variations on a single day are presented. Are the diurnal variations measured on a single day (which one?) or do they represent an average of several days? If latter, how many days were averaged and what are the standard deviations or errors of the means? If averages are presented – are their differences. i.e. the average diurnal variation statistically distinguishable and different for different periods?

Lines 500-504: Cai et al. (2007) mentions only a transport from upper level but not from stratosphere. Lelieveld et al. (2018), on the contrary, mentions a flux from the troposphere into the stratosphere in the region but not from stratosphere in the troposphere. Please refer correctly to cited literature.

Lines 506-507: "Atmosphere Hg has been reported to have strongly declined. . ." reads as a universal downward trend. That is generally not true – the downward trend has been observed only in North America and Europe in the last 10 – 20 years. Hg concentrations decreased in the southern hemisphere between 1996 and 2004, increased between 2007 and 2012 and remained nearly constant since. The records for East Asia are mostly too short to allow a general statement – see also the cited work by Tang et al. (2018). In this discussion, I would recommend to use emission inventories and their temporal change instead of trends Hg concentrations.

Editorial comments

Line 42: Why "unexpectedly"? Increase of GOM concentrations with altitude is predicted by some models and evidenced by observations such as at Mount Bachelor.

Line 62-63: The term "half-life" is unusual in atmospheric chemistry. "Lifetime" is usually used and clearly defined. A lifetime of 1- 2 years is somewhat long, current global models estimate GEM lifetime as short as several months. Please add references.

Line 80: "invasions" reads like a military term, "flux" or "import" may be more appropriate.

Lines 90-92: "The ..Hg concentrations. . . originated from. . ." is incorrect because as a consequence of the long GEM lifetime nobody can say where Hg came from. "The air masses carrying high Hg concentrations originated or, better; passed over. . .." would sound more appropriate.

Lines 120-122: "This monitoring site.."repeats a statement in lines 96-97. One of these statements is redundant.

Line 124: Why "comprehensive" when GEM, GOM and PBM are listed?

Line 254: "significantly" – at which level of significance?

Line 542-543: "air masses passed over Himalaya" is more credible than "air masses passed through Himalaya".

Lines 566-567: "Atmos." Instead of "Atoms." Dtto lines 560, 572, 588, 606, 608, 734, etc. Page numbers?

Figure captions contain generally too few information about what the figures display. A figure with its caption should be understandable without reading the paper.

Fig. 3: Solar radiation is difficult to discern, please correct.

Fig. 4: What are the units of wind speed? Please add to the figure or state in the figure

caption.

Fig.5: It would be desirable if the caption contained some information about what the authors understand under "back trajectories analysis".

Fig. 6 - caption: What "concepts" are shown by the maps?

Fig S2 – caption: What do the diagrams show? Presumably averages, medians, some percentiles – but what is what?

Figure S3: The capture states "Changes of snow cover rate and diurnal index..." Why rate when the y-axis is called snow coverage? What is the diurnal index? In both cases, the percents are of what?

Figure S4: The caption does not mention the diagram.

References

Ambrose, J.L.: Improved methods for signal processing in measurements of mercury by Tekran 2537A and 2537B instruments, Atmos. Meas. Tech., 10, 5063-5073, 2017.

Slemr, F., Weigelt, A., Ebinghaus, R., Kock, H.H., Bödewadt, J., Brenninkmeijer, C.A.M., Rauthe-Schöch, A., Weber, S., Hermann, M., Becker, J., Zahn, A., and Martinsson, B.: Atmospheric mercury measurements onboard the CARIBIC passenger aircraft, Atmos. Meas. Tech., 9, 2291-2302, 2016.

Sprovieri, F., Pirrone, N., Ebinghaus, R., Kock, H., and Dommergue, A.: A review of worldwide atmospheric mercury measurements, Atmos. Chem. Phys., 10, 8245-8265, 2010.

Sprovieri, F., et al.: Atmospheric mercury observations observed at ground-based monitoring sites globally distributed in the framework of the GMOS network, Atmos. Chem. Phys., 16, 11915-11935, 2016.

---

## Referee Comment (RC2) · Anonymous Referee #1 · 8 Oct 2018

This manuscript by Huiming Lin et al. presents the first record of atmospheric mercury species (GEM, GOM, PBM) during the Indian monsoon transition period in the Qomolangma Nature Preserve, located at the southern edge of the Tibetan Plateau along the border with the Indian subcontinent. Higher GEM concentrations during the monsoon period are attributed to air masses originating from east Nepal and Bangladesh, i.e. transboundary transport of Hg. Given the projected increase in Hg emissions in South and South-East Asia, monitoring data from downwind remote sites are essential. I think that this manuscript could make a valuable addition to the literature. However,

and in agreement with reviewer #2, I strongly suggest an update of the references list (imprecise citations throughout the manuscript) along with other edits (see below).

Lines 38-40 (and throughout the manuscript): Could you please add standard deviations every time you refer to a mean concentration? Additionally, did you perform a statistical test to demonstrate that there is indeed a significant difference between ISM and non-ISM concentrations?

Lines 42-44: I don't think that GOM concentrations of $\sim$ 20 pg/m3 are "considerably" higher than values in other clean or polluted regions. Concentrations of 1-20 pg/m3 are often reported at background/remote sites (e.g., Sprovieri et al. 2016) while hundreds of pg/m3 have been reported at urban/polluted sites (e.g., Duan et al. 2017; Han et al. 2018; Guo et al. 2017; Das et al. 2016).

Lines 49-52: To me, GEM concentrations reported in this study are at the lower end of concentrations reported in the Northern Hemisphere (Sprovieri et al. 2016). However, I do agree that international cooperation to limit Hg emissions is of utmost importance.

Line 61: I think reference to a review paper on Hg chemistry and atmospheric cycle is more appropriate here (e.g., Selin 2009).

Line 63: Recent modeling studies suggest a shorter lifetime in the atmosphere: 0.3-1 year (Selin 2009; Horowitz et al. 2017).

Line 64: Again, reference to Fang et al., 2009 is not appropriate here. Cite the original paper or a review paper.

Lines 73-74: Add Sprovieri et al. (2016) here.

Line 110: You could also briefly discuss future projections here (e.g., Pacyna et al. 2016).

Lines 120-122: I agree that this is the first study in the QNNP, but not the first one on the impact of the monsoon on Hg concentrations in Asia (e.g., Sheu et al. 2010; Yin

et al. 2018; Wang et al. 2018; Zhang et al. 2014, 2016). This should be more clearly stated.

Section 2.2: What is the time resolution of GEM measurements (e.g., 5 or 15 minutes)? If 5 minutes, concentrations are most likely biased low and should be adjusted upwards (Slemr et al. 2016; Ambrose 2017).

Line 202: Why did you use an arrival height of 1500 m a.g.l.? According to lines 159-161, the height of the boundary layer is ∼2000 m during the day and ∼ 350 m at night. This means that your back trajectories are well within the convective boundary layer during the day, but above the nocturnal boundary layer. Surface measurements at night are likely decoupled from what is happening in the residual layer and have a fairly restricted footprint. I am worried that these night-time trajectories may not be a good indication of source regions, especially given the complexity of the site. It is of common practice to use a height of 0.5 PBL.

Lines 254-258: I agree with the overall PBM decrease but you should perhaps add a sentence here saying that higher PBM concentrations during ISM2 will be addressed later in the manuscript (Section 3.3.2).

Line 261: Add Sprovieri et al. (2016) here.

Lines 275-277: See previous comment; GOM concentrations are "at the upper end of" (and not "much higher than") values in clean regions and are not higher than concentrations reported in polluted regions (e.g., Duan et al. 2017; Han et al. 2018; Guo et al. 2017; Das et al. 2016).

Lines 307-311: Please add standard deviations. I would like to see something like the 95 % confidence interval for the mean on Figure 3.

Line 343: Add here what's written lines 159-161 ("the height of the atmospheric boundary layer changes significantly in one day from ∼350 m above ground level during the night to ∼2000 m during the day").

Lines 344-363: I am not really convinced by the arguments here. Do you expect higher GEM concentrations in the afternoon to be due to local emissions? Have you checked whether you have such an increase every day, i.e., no wind direction influence? Or more or less emissions under more or less radiation? You seem to have all the data needed to perform a more thorough analysis. Could it be due to the boundary layer height? Is the boundary layer lower during the monsoon period? Is there any correlation with radiation or temperature? You could perhaps investigate the correlation between delta-GEM and delta-temperature or something like that.

Lines 378-388: In Figure 4, could you please use something else than shades of green. It is hard to tell the difference between <1.5 and >1.5 ng/m3.

Lines 410-412: How can you explain that GEM concentrations in air masses originating from the Tibetan Plateau were the highest?

Lines 415-417: "The clusters were similar to most of the clusters during the PISM period; however, the GEM concentrations in these clusters were higher than those during the PISM period". Could you explain why?

Lines 452-454: What about Bangladesh? Additionally, you don't really explain why GEM concentrations increase during the ISM period.

Line 464: Could you please add the dates for ISM2 here and/or add ISM2 in Figure S3?

Lines 464-466: Large amounts of PBM "may have been released". In this section and throughout the manuscript, please use the conditional tense to express conjectures/hypotheses.

Line 471: The discussion is about PBM here, not GOM. Remove reference to GOM.

Line 478: Can you explain this high value? Where did the air masses come from?

Lines 484: As mentioned above, 1.3 ng/m3 is at the low end of GEM concentrations

reported in the Northern Hemisphere. I agree that there is indeed an influence from South Asia, but concentrations on the QNNP are still fairly low during the PISM. I feel like you should slightly nuance your position.

Lines 487-495: Could you possibly add a comparison between PISM and ISM periods in Figure 7? This comparison is the core of your manuscript.

Line 503: "significant" rather than "considerable".

Line 507: Not true everywhere (e.g., Martin et al. 2017).

Line 516: Do you know if India, Nepal and Bangladesh have ratified the Minamata Convention on Hg? Check here: http://mercuryconvention.org/Countries/Parties/tabid/3428/language/en-US/Default.aspx. Hg emissions are projected to increase in India (Pacyna et al. 2016), what about Nepal and Bangladesh? You can perhaps strengthen the discussion here.

Lines 526-528: Is there a significant difference?

Lines 544-546: Again, concentrations reported here during PISM are at the low end of concentrations reported in the Northern Hemisphere. Additionally, concentrations are similar to those recently reported at Nam Co station on the Tibetan Plateau (Yin et al. 2018).

Figure 1: I assume that the red star within the QNNP is the location of the monitoring station. What about the two other red stars (Lhasa and Xigaze)? Do they represent cities and potential emissions? You should perhaps use a different type of star (monitoring site vs. cities) and make it clear in the caption.

Figure 2: Could you please add on this Figure the different periods (ISM1-5) you're referring to in Table 1?

Figure 3: I can't read the yellow axis, it is too bright. Please use another color. Additionally, what do you mean by GOM or PBM? Is this GOM, PBM, or the sum of the two? It is hard to see the dots and the diurnal cycle for GOM/PBM.

Figure 4: Which one is GOM, which one is PBM? Add a), b), c) on the Figure and caption.

Figure 6: Could you please explain in the caption what these values are? Probability of air passes passing through these regions?

Table 2: I think you can focus on Asian sites or refer to Figure 1 in Yin et al. (2018). The concentration reported for Nam Co station is incorrect (Yin et al. 2018).

Figure S4: Could you please add PISM, ISM1-5? Additionally, instead of April-August, is it possible to plot fires during PISM, ISM1-5? It would make it easier to identify whether fires are indeed more frequent in the area of interest during ISM2.

References:

Ambrose, J. L. 2017. "Improved Methods for Signal Processing in Measurements of Mercury by Tekran® 2537A and 2537B Instruments." Atmos. Meas. Tech. 10 (12): 5063–73. https://doi.org/10.5194/amt-10-5063-2017.

Das, Reshmi, Xianfeng Wang, Bahareh Khezri, Richard D. Webster, Pradip Kumar Sikdar, and Subhajit Datta. 2016. "Mercury Isotopes of Atmospheric Particle Bound Mercury for Source Apportionment Study in Urban Kolkata, India." Elem Sci Anth 4 (0): 000098. https://doi.org/10.12952/journal.elementa.000098.

Duan, Lian, Xiaohao Wang, Dongfang Wang, Yusen Duan, Na Cheng, and Guangli Xiu. 2017. "Atmospheric Mercury Speciation in Shanghai, China." Science of The Total Environment 578 (February): 460–68. https://doi.org/10.1016/j.scitotenv.2016.10.209.

Guo, Junming, Shichang Kang, Jie Huang, Qianggong Zhang, Maheswar Rupakheti, Shiwei Sun, Lekhendra Tripathee, et al. 2017. "Characterizations of Atmospheric Particulate-Bound Mercury in the Kathmandu Valley of Nepal,

[Figure]

South Asia." Science of The Total Environment 579 (February): 1240–48. https://doi.org/10.1016/j.scitotenv.2016.11.110.

Han, Deming, Jiaqi Zhang, Zihao Hu, Yingge Ma, Yusen Duan, Yan Han, Xiaojia Chen, Yong Zhou, Jinping Cheng, and Wenhua Wang. 2018. "Particulate Mercury in Ambient Air in Shanghai, China: Size-Specific Distribution, Gas–Particle Partitioning, and Association with Carbonaceous Composition." Environmental Pollution 238 (July): 543–53. https://doi.org/10.1016/j.envpol.2018.03.088.

Horowitz, H. M., D. J. Jacob, Y. Zhang, T. S. Dibble, F. Slemr, H. M. Amos, J. A. Schmidt, E. S. Corbitt, E. A. Marais, and E. M. Sunderland. 2017. "A New Mechanism for Atmospheric Mercury Redox Chemistry: Implications for the Global Mercury Budget." Atmos. Chem. Phys. 17 (10): 6353–71. https://doi.org/10.5194/acp-17-6353-2017.

Martin, L. G., C. Labuschagne, E.-G. Brunke, A. Weigelt, R. Ebinghaus, and F. Slemr. 2017. "Trend of Atmospheric Mercury Concentrations at Cape Point for 1995–2004 and since 2007." Atmos. Chem. Phys. 17 (3): 2393–99. https://doi.org/10.5194/acp-17-2393-2017.

Pacyna, J. M., O. Travnikov, F. De Simone, I. M. Hedgecock, K. Sundseth, E. G. Pacyna, F. Steenhuisen, N. Pirrone, J. Munthe, and K. Kindbom. 2016. "Current and Future Levels of Mercury Atmospheric Pollution on a Global Scale." Atmos. Chem. Phys. 16 (19): 12495–511. https://doi.org/10.5194/acp-16-12495-2016.

Selin, Noelle E. 2009. "Global Biogeochemical Cycling of Mercury: A Review." Annual Review of Environment and Resources 34 (1): 43–63. https://doi.org/10.1146/annurev.environ.051308.084314.

Sheu, G.-R., N.-H. Lin, J.-L. Wang, C.-T. Lee, C.-F. O. Yang, and S.-H. Wang. 2010. "Temporal Distribution and Potential Sources of Atmospheric Mercury Measured at a High-Elevation Background Station in Taiwan." Atmospheric Environment 44: 2393–2400.

Slemr, F., A. Weigelt, R. Ebinghaus, H. H. Kock, J. Bödewadt, C. A. M. Brenninkmeijer, A. Rauthe-Schöch, et al. 2016. "Atmospheric Mercury Measurements Onboard the CARIBIC Passenger Aircraft." Atmos. Meas. Tech. 9 (5): 2291–2302. https://doi.org/10.5194/amt-9-2291-2016.

Sprovieri, F., N. Pirrone, M. Bencardino, F. D'Amore, F. Carbone, S. Cinnirella, V. Mannarino, et al. 2016. "Atmospheric Mercury Concentrations Observed at Ground-Based Monitoring Sites Globally Distributed in the Framework of the GMOS Network." Atmos. Chem. Phys. 16 (18): 11915–35. https://doi.org/10.5194/acp-16-11915-2016.

Wang, Xun, Che-Jen Lin, Xinbin Feng, Wei Yuan, Xuewu Fu, Hui Zhang, Qingru Wu, and Shuxiao Wang. 2018. "Assessment of Regional Mercury Deposition and Emission Outflow in Mainland China." Journal of Geophysical Research: Atmospheres 0 (ja). https://doi.org/10.1029/2018JD028350.

Yin, Xiufeng, Shichang Kang, Benjamin de Foy, Yaoming Ma, Yindong Tong, Wei Zhang, Xuejun Wang, Guoshuai Zhang, and Qianggong Zhang. 2018. "Multi-Year Monitoring of Atmospheric Total Gaseous Mercury at a Remote High-Altitude Site (Nam Co, 4730 M a.s.l.) in the Inland Tibetan Plateau Region." Atmospheric Chemistry and Physics 18 (14): 10557–74. https://doi.org/10.5194/acp-18-10557-2018.

Zhang, H., X. Fu, C.-J. Lin, L. Shang, Y. Zhang, X. Feng, and C. Lin. 2016. "Monsoon-Facilitated Characteristics and Transport of Atmospheric Mercury at a High-Altitude Background Site in Southwestern China." Atmos. Chem. Phys. 16 (20): 13131–48. https://doi.org/10.5194/acp-16-13131-2016.

Zhang, H., X. W. Fu, C.-J. Lin, X. Wang, and X. B. Feng. 2014. "Observation and Analysis of Speciated Atmospheric Mercury in Shangri-La, Tibetan Plateau, China." Atmospheric Chemistry and Physics.

---

## Author Comment (AC1) · 14 Dec 2018

**Responses to Reviewers' Comments**

**First Measurement of Atmospheric Mercury Species in Qomolangma Nature Preserve,**

**Tibetan Plateau, and Evidence of Transboundary Pollutant Invasion (acp-2018-806)**

Dear editor and reviewer,

We greatly appreciate the useful comments from the editor and reviewers. We think the novelty and importance of this study have been acknowledged by the reviewers. We have revised the original manuscript thoroughly based on the reviewers' comments. Detailed point by point responses are provided as follows. All the revisions have been highlighted in blue color in the manuscript. We hope the revised manuscript could meet the standard of ACP. Thanks again for your considerations.

**Anonymous Referee #2**

**General comment**

The authors present speciated Hg measurements (GEM, GOM, and PBM) at a high altitude station in Tibet near the border to Nepal. They show a pronounced concentration differences between pre-monsoon and monsoon periods and explain them by changing transport patterns encompassing different source regions, especially those in Pakistan, India, and Bangladesh. They also show influence of biomass burning. There are only a few measurements in this part of the world and, thus, they deserve to be published. Their interpretation is sound. Unfortunately, the data presentation is marred by at times awkward wording, imprecise citation of references, uninformative figure captions, etc., and thus it needs a good deal of editing. Some improvements are proposed below.

**Response**

Thanks for your comments and suggestions. We have polished the language of the manuscript, updated the citied references and revised the figure captions accordingly. Please see the revised manuscript. All the revisions have been highlighted in blue. Detailed responses to your comments are provided as follows.

**Specific comments**

**Comment #1**

Section 2.2: This section describes essentially the GOM and PBM measurement but not the measurement of GEM. Sampling time for GEM measurements has to be stated. The reason is that

the GEM (with usually 5 min sampling), GOM, and PBM data are probably biased low due to problems with the internal default integration because less than 10 pg was collected for the individual analysis (Slemr et al., 2016; Ambrose, 2017). This problem is especially important at the QNNP station because only flow rates of 0.75 and 7 l(STP) min$^{-1}$ were used for GEM and GOM/PBM measurements, respectively, instead of the usual 1 and 10 l(STP) min$^{-1}$. The authors should mention the bias and assess its average magnitude using Fig. 3 of Slemr et al. (2016). This is needed when the data are compared to measurements at other sites. A definition of standard pressure and temperature would be also helpful.

**Response #1**

Thanks for your suggestion. We agree with the reviewer that a small captured Hg amount would probably lead to the biases of the measurement in QNNP. According to the method by Slemr et al. (2016), the monitoring data with low captured Hg amounts (less than 10 pg) were recalculated. In this case, the monitoring data with GOM or PBM concentrations <23.8 pg m$^{-3}$ was recalculated. The revised average concentrations increase slightly from 21.3±13.5 pg m$^{-3}$ to 21.4±13.4 pg m$^{-3}$ for GOM, and from 25.5±19.2 pg m$^{-3}$ to 25.6±19.1 pg m$^{-3}$ for PBM, respectively. All the data have been updated in the revised manuscript. The GEM sampling time, a definition of standard pressure and temperature is also provided in the revised manuscript (Line 183, 186-187, 193-199 in the revised manuscript).

**Comment #2**

Section 2.4: The use of backward trajectories for identification of the source areas seems to me to be questionable in this particular case. If I understand it properly the trajectory arrival height was set 1500 above the station, i.e. at an altitude of some 5800 m. In addition, the station is located in a very complex terrain (mountains above 8000m) with local winds due to glacier coverage. The question is how well the trajectories are representative for the air analysed at the station? Can the authors say anything about it?

**Response #2**

We fully agree that the complex terrains and local glacial winds could affect the transport of the pollutants, which might cause biases between the real situation and simulated situation. To our knowledge, existing atmospheric Hg models are not able to address the impacts of local terrains (Gustin et al., 2015), which have been evidenced in many previous studies, such as Yin et al.

(2018)'s study in central Tibetan plateau, Zhang et al. (2016)'s study in southwestern China and Fu et al. (2012)'s study in the northeast Tibetan plateau. The local terrains in all these studies have not been addressed. As suggested by another reviewer, in the revised manuscript, we have reset the arrival height of air mass to be 1000 m a.g.l. to reflect the influence of boundary layers. We do appreciate the suggestion from the reviewer, and will explore to model the impacts of local terrains on atmospheric Hg transport in QNNP in the future.

As we discussed in section 3.3.2, during ISM2 period, the trajectories and potential source region analysis could well present the influence of biomass burning from north Indian. When the source regions have frequent biomass burning (fire hotspots), the GEM and PBM concentrations in QNNP would correspondingly increase. This may indicate that the trajectories can still well represent the air analysis under complex terrain in QNNP.

**Comment #3**

Section 3.1: Averages and standard deviations should always be given with the number of measurements since only with it the significance of the differences can be determined. Are the difference of GEM, GOM and PBM concentration between PISM and ISM periods statistically significant?

**Response #3**

We have provided the number of measurements and statistical information in the revised manuscript. Please see the revised Section 3.1. Thanks for your suggestion.

**Comment #4**

Lines 278-283: Subsidence is probably only a part of the explanation; lack of precipitation could be another part.

**Response #4**

We agree that rare precipitation in QNNP could be an important reason for the high GOM in this region. As stated in the section of Methods and materials, the annual precipitation in QNNP is only 270.5 mm (Chen et al., 2016). We have provided the following information in the revised manuscript, "**Low wet deposition of GOM caused by rare precipitation in QNNP (~270mm) (Chen et al., 2016) could be another reason for the high GOM concentration (Prestbo and Gay, 2009)**". (Line 307-309 in the revised manuscript).

**Comment #5**

Table 2 claims to summarize global measurements of GEM, GOM, and PBM which is far from being true. Outside of Asia only three US sites are listed which is only a small fraction of all measurements (Sprovieri et al., 2010, 2017; Gay et al., 2013). In addition, these three US sites are not mentioned in the text. Since a comprehensive list would fill several pages I would recommend to concentrate on the measurements in Asia and for comparison with worldwide concentrations only to refer to above references.

**Response #5**

Thanks for your suggestions. Yes, we agree that it would be better to focus on the atmospheric Hg monitoring in Asia. In the revised manuscript, we have removed the monitoring sites outside of Asia from the Table 2. Please also see the revised manuscript (Line 279-280).

**Comment #6**

Section 3.2: In the text a sum of GOM and PBM is discussed but in the legend of Figure 3 symbols are declared as PBM or GOM. Please correct. The caption of Figure 3 reads as if the presented diurnal variations were representative of different periods, i.e. as averages of several days, but the reader has an impression that diurnal variations on a single day are presented. Are the diurnal variations measured on a single day (which one?) or do they represent an average of several days? If latter, how many days were averaged and what are the standard deviations or errors of the means? If averages are presented – are their differences. i.e. the average diurnal variation statistically distinguishable and different for different periods?

**Response #6**

Thanks for your suggestion.

//In the original Figure 3, GOM and PBM were displayed by using hollow and solid blue dots, respectively. We have added a new label to make it clear for readers.

//The data presented in Figure 3 is the average value of the monitoring data in each period (PISM, ISM 1-5), and this has been clarified in the caption of revised Figure 3. Number of days to calculate the average in each period is also provided. Please see the revised Figure 3.

//We agree that it would be better to provide standard deviations of different monitoring data in Figure 3. However, there are many colored lines in the original Figure 3. Hence, we have added a Figure S3 in the revised manuscript to describe the uncertainty in atmospheric Hg monitoring data. Please see Figure S3 in the revised manuscript.

**Comment #7**

Lines 500-504: Cai et al. (2007) mentions only a transport from upper level but not from stratosphere. Lelieveld et al. (2018), on the contrary, mentions a flux from the troposphere into the stratosphere in the region but not from stratosphere in the troposphere. Please refer correctly to cited literature.

**Response #7**

Thanks for your suggestion. We have revised this sentence as follows: "**As showed in other studies in the northern or eastern Tibetan Plateau, the glacier wind can pump down air masses from upper level to the surface in QNNP (Cai et al., 2007). The pump movement is remarkably efficient at transporting air masses (Zhu et al., 2006), and could bring significant amount of pollutants to QNNP.**" (Line 536-540 in the revised manuscript).

**Comment #8**

Lines 506-507: "Atmosphere Hg has been reported to have strongly declined. . ." reads as a universal downward trend. That is generally not true – the downward trend has been observed only in North America and Europe in the last 10 – 20 years. Hg concentrations decreased in the southern hemisphere between 1996 and 2004, increased between 2007 and 2012 and remained nearly constant since. The records for East Asia are mostly too short to allow a general statement – see also the cited work by Tang et al. (2018). In this discussion, I would recommend to use emission inventories and their temporal change instead of trends Hg concentrations.

**Response #8**

Thanks for your suggestions. We agree with the reviewer that the downward trend of atmospheric Hg concentrations was only observed in North America and Europe (Gay et al., 2013;Sprovieri et al., 2016). In 2016, we published a paper to describe the changes of atmospheric Hg between 2006-2015 in Tibet (Tong et al., 2016). Through the analysis of leaves of *Androsace tapete* that represent growing periods spanning the past decade, we found that there was a significant decrease of atmospheric Hg since 2010 in Tibet. Based on the reviewer's suggestion, in the revised manuscript, we have provided the description about historical change of atmospheric Hg emissions in China, as follows: "**According to the recently updated emission inventory in China (Wu et al., 2016), anthropogenic Hg emissions in China reached a peak amount of about 567 tonnes in 2011 and have decreased since then. In 2014, the anthropogenic Hg**

**emissions decreased to 530 tonnes. This was also confirmed the concentration of plant Hg from a sampling site near QNNP, which recorded the decrease of atmospheric Hg concentration in Tibet since the year of 2010 (Tong et al., 2016)**." (Please see Line 548-553 in the revised manuscript).

**Comments #9**

Line 42: Why "unexpectedly"? Increase of GOM concentrations with altitude is predicted by some models and evidenced by observations such as at Mount Bachelor.

**Response #9**

We have deleted this word accordingly.

**Comments #10**

Line 62-63: The term "half-life" is unusual in atmospheric chemistry. "Lifetime" is usually used and clearly defined. A lifetime of 1- 2 years is somewhat long, current global models estimate GEM lifetime as short as several months. Please add references.

**Response #10**

//We have replaced "half-life" with "lifetime" in the revised manuscript.

//We have updated the information of GEM lifetime. After reviewing previous studies (Selin, 2009;Horowitz et al., 2017;Travnikov et al., 2017), we think ~0.3-1 year might be appropriate. Please see Line 63-66 in the revised manuscript.

**Comment #11**

Line 80: "invasions" reads like a military term, "flux" or "import" may be more appropriate.

**Response #11**

We have replaced the word with "import" accordingly. Thanks for the suggestion.

**Comment #12**

Lines 90-92: "The. Hg concentrations. . . originated from. . ." is incorrect because as a consequence of the long GEM lifetime nobody can say where Hg came from. "The air masses carrying high Hg concentrations originated or, better; passed over. . .." would sound more appropriate.

**Response #12**

We have revised the sentence as follows: "**Fu et al. (2012a) report that air masses with high Hg concentrations passed over the urban and industrial areas in Western China and**

**Northern India, and influenced the atmospheric Hg concentrations in Waliguan on the northeastern edge of the Tibetan Plateau.**" (Line 92-95 in the revised manuscript).

**Comment #13**

Lines 120-122: "This monitoring site.." repeats a statement in lines 96-97. One of these statements is redundant.

**Response #13**

We have deleted this sentence from the manuscript.

**Comment #14**

Line 124: Why "comprehensive" when GEM, GOM and PBM are listed?

**Response #14**

We have deleted this word accordingly

**Comment #15**

Line 254: "significantly" – at which level of significance?

**Response #15**

This sentence has been revised as follows: "**Figure S2 shows that GEM concentrations increased significantly with the development of ISM (p<0.001 between ISM1 and ISM4), while decreases of GOM and PBM concentrations were observed during the study period (p<0.001, between ISM1 and ISM5), with decreases of 37.9% (from 20.3±7.38 pg m$^{-3}$ to 12.6 ±8.82 pg m$^{-3}$) and 48.1% (from 21.2±7.38 pg m$^{-3}$ to 11.0±5.85 pg m$^{-3}$), respectively**". Please see Line 272-277 in the revised manuscript.

**Comment #16**

Line 542-543: "air masses passed over Himalaya" is more credible than "air masses passed through Himalaya".

**Response #16**

We have replaced it accordingly (Line 587-589 in the revised manuscript).

**Comment #17**

Lines 566-567: "Atmos." Instead of "Atoms." Dtto lines 560, 572, 588, 606, 608, 734, etc. Page numbers?

**Response #17**

We have replaced this word and the whole manuscript has been checked and revisions have

been made.

**Comment #18**

Figure captions contain generally too few information about what the figures display. A figure with its caption should be understandable without reading the paper.

Response #18

We have updated the figure captions in the revised manuscript, as follows:

"**Figure 1. Location of Qomolangma National Nature Preserve (QNNP). The red star shows the location of the monitoring station in QNNP. The red dots show the locations of two largest cities in Tibet (Lhasa and Xigaze), with the scale bars showing their distances from the QNNP.**

**Figure 2. Time series change of GEM, GOM and PBM concentration during the study period. The time series was split into a Pre-Indian Summer Monsoon (PISM) period (15 April-30 April, 2016) and 5 Indian Summer Monsoon (ISM) periods (1 May–12 May (ISM1), 13 May–4 June (ISM2), 5 June–20 June (ISM3), 21 June–10 July (ISM4), 11 July–14 August (ISM5)).**

**Figure 3. Diurnal variations of GEM, GOM and PBM concentrations during the Pre-Indian Summer Monsoon (PISM) period (15 April–30 April, 2016) and 5 Indian Summer Monsoon (ISM) periods (1 May–12 May (ISM1), 13 May–4 June (ISM2), 5 June–20 June (ISM3), 21 June–10 July (ISM4), 11 July–14 August (ISM5)). The concentrations represent the daily average values during each period.**

**Figure 4. Concentration roses of GEM, GOM and PBM from different wind directions. The length of each spoke describes the frequency of flow from the corresponding direction.**

**Figure 5. Clusters of the Back trajectories analysis from the Qomolangma National Nature Preserve (QNNP) monitoring site during the Pre-Indian Summer Monsoon (PISM) period and the 5 Indian Summer Monsoon (ISM) periods. The cluster statistics summarize the percentage of back trajectories for each cluster. The background color shading represents the global Hg emissions from anthropogenic sources (UNEP, 2013).**

**Figure 6. Potential source regions and pathways of GEM using the Potential Source Contribution Function (PSCF) method before and during the Indian Summer Monsoon (ISM). PSCF values represent the probability that a grid cell is a source of Hg.**

**Figure 7. Conceptual map of transboundary transport of atmospheric Hg in the Himalaya region. Arrows show the impacts of the Indian Summer Monsoon, upward winds and glacial winds on the transboundary transport of Hg.**

"

**Comment #19**

Fig. 3: Solar radiation is difficult to discern, please correct.

**Response #19**

We have regulated the color of solar radiation in Figure 3, and please see the revised figure.

**Comment #20**

Fig. 4: What are the units of wind speed? Please add to the figure or state in the figure caption.

**Response #20**

Fig. 4 describes the frequency and concentration distribution of atmospheric Hg at different wind directions. The length of each spoke describes the frequency of atmospheric Hg concentration at certain wind direction. So, this value is irrelevant with the wind speeds.

**Comment #21**

Fig.5: It would be desirable if the caption contained some information about what the authors understand under "back trajectories analysis".

**Response #21**

The figure caption has been revised as follows: "**Figure 5. Clusters of the Back trajectories analysis from the Qomolangma National Nature Preserve (QNNP) monitoring site during the Pre-Indian Summer Monsoon (PISM) period and the 5 Indian Summer Monsoon (ISM) periods. The cluster statistics summarize the percentage of back trajectories for each cluster. The background color shading represents the global Hg emissions from anthropogenic sources (UNEP, 2013).**" Please see Line 899-904 in the revised manuscript. All the figure captions in the manuscript have been revised.

**Comment #22**

Fig. 6 - caption: What "concepts" are shown by the maps?

**Response #22**

The caption of Fig. 6 has been revised as follows: "**Figure 6. Potential source regions and pathways of GEM using the Potential Source Contribution Function (PSCF) method before**

**and during the Indian Summer Monsoon (ISM). PSCF values represent the probability that a grid cell is a source of Hg.**"

**Comment #23**

Fig S2 – caption: What do the diagrams show? Presumably averages, medians, some percentiles – but what is what?

**Response #23**

We have added a legend in the revised Fig S2. Please see the revised manuscript.

**Comment #24**

Figure S3: The capture states "Changes of snow cover rate and diurnal index..." Why rate when the y-axis is called snow coverage? What is the diurnal index? In both cases, the percents are of what?

**Response #24**

We have replaced the "snow cover rate" with "snow coverage percentage" in the revised manuscript. To avoid the misunderstanding, we have deleted the diurnal index in the revised figure.

**Comments #25**

Figure S4: The caption does not mention the diagram.

**Response #25**

The figure caption has been revised as follows: "**Changes of snow coverage in QNNP during the study period (data from MODIS, MOD10A1)**" Please see the revised Figure S5.

**Reference**

Chen, P., Gao, Y., Lee, A. T., Cering, L., Shi, K., and Clark, S. G.: Human–carnivore coexistence in Qomolangma (Mt. Everest) Nature Reserve, China: Patterns and compensation, Biological conservation, 197, 18-26, 2016.

Fu, X., Feng, X., Liang, P., Zhang, H., Ji, J., and Liu, P.: Temporal trend and sources of speciated atmospheric mercury at Waliguan GAW station, Northwestern China, Atmospheric Chemistry and Physics, 12, 1951-1964, 2012.

Gay, D. A., Schmeltz, D., Prestbo, E., Olson, M., Sharac, T., and Tordon, R.: The Atmospheric Mercury Network: measurement and initial examination of an ongoing atmospheric mercury record across North America, Atmospheric Chemistry and Physics, 13, 11339-11349,'D'O'I:' 10.5194/acp-13-11339-2013, 2013.

Gustin, M. S., Amos, H. M., Huang, J., Miller, M. B., and Heidecorn, K.: Measuring and modeling mercury in the atmosphere: a critical review, Atmospheric Chemistry and Physics, 15, 5697-5713, 2015.

Horowitz, H. M., Jacob, D. J., Zhang, Y., Dibble, T. S., Slemr, F., Amos, H. M., Schmidt, J. A., Corbitt, E. S., Marais, E. A., and Sunderland, E. M.: A new mechanism for atmospheric mercury redox chemistry: Implications for the global mercury budget, Atmospheric Chemistry and Physics, 17, 6353-6371, 2017.

Selin, N. E.: Global biogeochemical cycling of mercury: a review, Annual Review of Environment and Resources, 34, 43-63, 2009.

Slemr, F., Weigelt, A., Ebinghaus, R., Kock, H. H., Bödewadt, J., Brenninkmeijer, C. A., Rauthe-Schöch, A., Weber, S., Hermann, M., and Becker, J.: Atmospheric mercury measurements onboard the CARIBIC passenger aircraft, Atmospheric Measurement Techniques, 9, 2291-2302, 2016.

Sprovieri, F., Pirrone, N., Bencardino, M., D'Amore, F., Carbone, F., Cinnirella, S., Mannarino, V., Landis, M., Ebinghaus, R., and Weigelt, A.: Atmospheric mercury concentrations observed at ground-based monitoring sites globally distributed in the framework of the GMOS network, Atmospheric Chemistry and Physics, 16, 11915-11935, 2016.

Tong, Y., Yin, X., Lin, H., Wang, H., Deng, C., Chen, L., Li, J., Zhang, W., Schauer, J. J., and Kang, S.: Recent Decline of Atmospheric Mercury Recorded by Androsace tapete on the Tibetan Plateau, Environmental science & technology, 50, 13224-13231, 2016.

Travnikov, O., Angot, H., Artaxo, P., Bencardino, M., Bieser, J., D'Amore, F., Dastoor, A., Simone, F. D., Diéguez, M. d. C., and Dommergue, A.: Multi-model study of mercury dispersion in the atmosphere: atmospheric processes and model evaluation, Atmospheric Chemistry and Physics, 17, 5271-5295, 2017.

Wu, Q., Wang, S., Li, G., Liang, S., Lin, C.-J., Wang, Y., Cai, S., Liu, K., and Hao, J.: Temporal trend and spatial distribution of speciated atmospheric mercury emissions in China during 1978–2014, Environmental science & technology, 50, 13428-13435, 2016.

Yin, X., Kang, S., Foy, B. d., Ma, Y., Tong, Y., Zhang, W., Wang, X., Zhang, G., and Zhang, Q.: Multi-year monitoring of atmospheric total gaseous mercury at a remote high-altitude site (Nam Co, 4730 m asl) in the inland Tibetan Plateau region, Atmospheric Chemistry and Physics, 18, 10557-10574, 2018.

Zhang, H., Fu, X., Lin, C., Wang, X., and Feng, X.: Observation and analysis of speciated atmospheric mercury in Shangri-La, Tibetan Plateau, China, Atmos. Chem. Phys, 15, 653-665,

2015.

Zhang, H., Fu, X., Lin, C.-J., Shang, L., Zhang, Y., Feng, X., and Lin, C.: Monsoon-facilitated characteristics and transport of atmospheric mercury at a high-altitude background site in southwestern China, Atmospheric Chemistry & Physics, 16, 2016.

---

## Author Comment (AC2) · 14 Dec 2018

**Responses to Reviewers' Comments**

**First Measurement of Atmospheric Mercury Species in Qomolangma Nature Preserve,**

**Tibetan Plateau, and Evidence of Transboundary Pollutant Invasion (acp-2018-806)**

Dear editor and reviewer,

We greatly appreciate the useful comments from the editor and reviewers. We think the novelty and importance of this study have been acknowledged by the reviewers. We have revised the original manuscript thoroughly based on the reviewers' comments. Detailed point by point responses are provided as follows. All the revisions have been highlighted in blue in the manuscript. We hope the revised manuscript could meet the standard of ACP. Thanks again for your considerations.

**Anonymous Referee #1**

**General comment**

This manuscript by Huiming Lin et al. presents the first record of atmospheric mercury species (GEM, GOM, PBM) during the Indian monsoon transition period in the Qomolangma Nature Preserve, located at the southern edge of the Tibetan Plateau along the border with the Indian subcontinent. Higher GEM concentrations during the monsoon period are attributed to air masses originating from east Nepal and Bangladesh, i.e. transboundary transport of Hg. Given the projected increase in Hg emissions in South and South-East Asia, monitoring data from downwind remote sites are essential. I think that this manuscript could make a valuable addition to the literature. However, and in agreement with reviewer #2, I strongly suggest an update of the references list (imprecise citations throughout the manuscript) along with other edits (see below).

**Response**

Thanks for the helpful comments and suggestions. We have updated the reference list and addressed other concerns from the reviewer in the revised manuscript. A detailed point by point responses to the comments have been provided as follows.

**Specific comments**

**Comment #1**

Lines 38-40 (and throughout the manuscript): Could you please add standard deviations every time you refer to a mean concentration? Additionally, did you perform a statistical test to demonstrate that there is indeed a significant difference between ISM and non-ISM

concentrations?

**Response #1**

In the revised manuscript, we have added the standard deviations with the mean concentrations, and statistical test results have been added throughout the manuscript when necessary. The GEM concentrations in the ISM period were significantly higher than that in the PISM period, the GOM and PBM concentrations in the ISM period were significantly lower than those in the PISM period ($p < 0.001$, ANOVA test). We have also checked the whole manuscript and added the statistical results when necessary. Please see the revised manuscript.

**Comment #2**

Lines 42-44: I don't think that GOM concentrations of ~20 pg/m$^3$ are "considerably" higher than values in other clean or polluted regions. Concentrations of 1-20 pg/m$^3$ are often reported at background/remote sites (e.g., Sprovieri et al. 2016) while hundreds of pg/m$^3$ have been reported at urban/polluted sites (e.g., Duan et al. 2017; Han et al. 2018; Guo et al. 2017; Das et al. 2016).

**Response #2**

Thanks for the comment. We totally agree that there are some monitoring sites with higher GOM concentrations than the measured values in QNNP. However, if we compared the GOM concentrations ($35.2 \pm 18.6$ pg m$^{-3}$ during PISM period and $19.3 \pm 10.9$ pg m$^{-3}$ during ISM period) in QNNP with the monitored values from other monitoring sites in China, we found the values in QNNP were still high considering its low GEM concentrations (as shown in Table 2). For instance, the reported GOM concentrations in Beijing and Shanghai, which have been polluted by quick industrial development for a long time, were $10.1 \pm 18.8$ and $21 \pm 100$ pg m$^{-3}$ (Zhang et al., 2013;Duan et al., 2017). In the background monitoring sites such as Waliguan (Fu et al., 2012) and Ailaoshan (Zhang et al., 2016), the measured GOM concentrations were $7.4 \pm 4.8$ pg m$^{-3}$ and $2.3 \pm 2.3$ pg m$^{-3}$, respectively. However, we acknowledge that the word "considerably" could cause misunderstanding by readers, and we have revised this sentence as follows: "**Relative to the low GEM concentrations, GOM concentrations (with a mean value of $21.3 \pm 13.5$ pg m$^{-3}$) in this region were relatively high compared with the measured values in some other regions of China.**" (Line 42-45 in the revised manuscript).

**Comment #3**

Lines 49-52: To me, GEM concentrations reported in this study are at the lower end of concentrations reported in the Northern Hemisphere (Sprovieri et al. 2016). However, I do agree that international cooperation to limit Hg emissions is of utmost importance.

**Response #3**

We agree that, in general, the GEM concentrations in QNNP are relatively low compared with other monitored values in Northern Hemisphere (Wan et al., 2009;Fu et al., 2012;Sprovieri et al.,

2016). From our study, we found that the atmospheric GEM concentrations could increase significantly from the PISM period ($1.31\pm0.42$ ng m$^{-3}$) to the ISM period ($1.44\pm0.36$ ng m$^{-3}$) in

QNNP (p<0.001). We have revised the sentence as follows: "**The atmospheric Hg concentration**

**in QNNP in the Indian Summer Monsoon period was significantly influenced by the**

**transboundary Hg flows. This sets forth the need for a more specific identification of Hg**

**sources impacting QNNP and underscores the importance of international cooperation for**

**global Hg controls.**" (Line 50-53 in the revised manuscript).

**Comment #4**

Line 61: I think reference to a review paper on Hg chemistry and atmospheric cycle is more appropriate here (e.g., Selin 2009).

**Response #4**

We have added the reference accordingly. Please see Line 60-62 in the revised manuscript.

**Comment #5**

Line 63: Recent modeling studies suggest a shorter lifetime in the atmosphere: 0.3-1 year (Selin

2009; Horowitz et al. 2017).

**Response #5**

We have updated the information about the lifetime of GEM in the revised manuscript. Please see Line 63-66 in the revised manuscript. Thanks for your suggestions.

**Comment #6**

Line 64: Again, reference to Fang et al., 2009 is not appropriate here. Cite the original paper or a review paper.

**Response #6**

We have deleted the reference in the revised manuscript. The following references are added:

(Selin, 2009;Horowitz et al., 2017;Travnikov et al., 2017).

**Comment #7**

Lines 73-74: Add Sprovieri et al. (2016) here.

**Response #7**

We have added this reference in the revised manuscript (Line 74-75).

**Comment #8**

Line 110: You could also briefly discuss future projections here (e.g., Pacyna et al. 2016).

**Response #8**

Thanks for your suggestions. We have reviewed some previous studies (Burger Chakraborty et al., 2013;Giang et al., 2015;Pacyna et al., 2016;Wu et al., 2018) and added more descriptions about the future atmospheric Hg emissions in China and India, as follows: "**China is predicted to**

**become the largest economy in the world in the next 20-50 years, and India is predicted to**

**catch up with the Euro area before 2030 (Pacyna et al., 2016). China is predicted to become**

**the largest economy in the world in the next 20-50 years, and India is predicted to catch up**

**with the Euro area before 2030 (Pacyna et al., 2016). With the implementation of control**

**strategies, the atmospheric Hg emissions is forecasted to be about 242 tonnes in China in**

**2020 (Wu et al., 2018). With the implementation of control strategies, the atmospheric Hg**

**emissions is forecasted to be about 242 tonnes in China in 2020 (Burger Chakraborty et al.,**

**2013).**" Please see Line 112-119 in the revised manuscript.

**Comment #9**

Lines 120-122: I agree that this is the first study in the QNNP, but not the first one on the impact of the monsoon on Hg concentrations in Asia (e.g., Sheu et al. 2010; Yin et al. 2018; Wang et al. 2018; Zhang et al. 2014, 2016). This should be more clearly stated.

**Response #9**

We have revised this sentence as follows: "**To the best of our knowledge, the present work is**

**the first study regarding Hg monitoring and source identification in the QNNP covering both**

**the period preceding the Indian Summer Monsoon (PISM) and during the Indian Summer**

**Monsoon (ISM).**" (Line 130-134 in the revised manuscript).

**Comment #10**

Section 2.2: What is the time resolution of GEM measurements (e.g., 5 or 15 minutes)? If 5

minutes, concentrations are most likely biased low and should be adjusted upwards (Slemr et al.

2016; Ambrose 2017).

**Response #10**

Thanks for your suggestion. In this study, the time resolution of GEM measurements is 5

minutes. We agree with the reviewer that the small captured Hg amount would probably cause the bias of the measurement. In the revised manuscript, the monitoring data with the low captured Hg (with a Hg amount lower than 10 pg) was adjusted based on the method of Slemr et al. (2016). All the data has been updated in the revised manuscript.

**Comment #11**

Line 202: Why did you use an arrival height of 1500 m a.g.l.? According to lines 159-161, the height of the boundary layer is ~2000 m during the day and ~ 350 m at night. This means that your back trajectories are well within the convective boundary layer during the day, but above the nocturnal boundary layer. Surface measurements at night are likely decoupled from what is happening in the residual layer and have a fairly restricted footprint. I am worried that these night-time trajectories may not be a good indication of source regions, especially given the complexity of the site. It is of common practice to use a height of 0.5 PBL.

**Response #11**

Thanks for your suggestion. According to your suggestions, we have reset the arrival height of air masses at 1000 m a.g.l. (0.5 PBL) in the revised manuscript. All simulations were recalculated according to the new arrival height of air masses. All the results have been updated in the revised manuscript. Please see the revised manuscript.

**Comment #12**

Lines 254-258: I agree with the overall PBM decrease but you should perhaps add a sentence here saying that higher PBM concentrations during ISM2 will be addressed later in the manuscript (Section 3.3.2).

**Response #12**

Thanks for the suggestion. The following sentence has been added into the revised manuscript, as follows: "**Reason for the higher PBM concentrations during ISM2 is discussed in Section**

**3.3.2.**". Please see Line 277-278 in the revised manuscript.

**Comment #13**

Line 261: Add Sprovieri et al. (2016) here.

**Response #13**

We have added the suggested reference in the revised manuscript (Line 283-284).

**Comment #14**

Lines 275-277: See previous comment; GOM concentrations are "at the upper end of" (and not

"much higher than") values in clean regions and are not higher than concentrations reported in polluted regions (e.g., Duan et al. 2017; Han et al. 2018; Guo et al. 2017; Das et al. 2016).

**Response #14**

We agree with the reviewer that "much higher than" may cause misunderstanding to the readers, and we have revised it in the manuscript, as follows: "**However, despite its low GEM**

**concentration, GOM concentration (with a value of 21.4±13.4 pg m$^{-3}$) in QNNP was**

**relatively high compared with the values in the clean regions (usually lower than 10 pg m$^{-3}$,**

**Table 2) or even some polluted regions of China (such as the suburban area of Beijing**

**(10.1±18.8 pg m$^{-3}$), Shanghai (21±100 pg m$^{-3}$)(Zhang et al., 2013;Duan et al., 2017)) (Table**

**2).** " (Line 297-301 in the revised manuscript).

**Comment #15**

Lines 307-311: Please add standard deviations. I would like to see something like the 95 %

confidence interval for the mean on Figure 3.

**Response #15**

We have provided the standard divisions for all the mean values throughout the manuscript. We have added another figure in the revised manuscript (Figure S3) and 95% CI has been added, since too many colored lines are in the original Figure 3, and they are difficult to identify.

**Comment #16**

Line 343: Add here what's written lines 159-161 ("the height of the atmospheric boundary layer changes significantly in one day from ~350 m above ground level during the night to ~2000 m during the day").

**Response #16**

We have added the following sentence in Line 363-365 of the revised manuscript: "**The height**

**of the atmospheric boundary layer could vary significantly, from ~350 m above ground level**

**to ~2000 m in one day.**"

**Comment #17**

Lines 344-363: I am not really convinced by the arguments here. Do you expect higher GEM

concentrations in the afternoon to be due to local emissions? Have you checked whether you have such an increase every day, i.e., no wind direction influence? Or more or less emissions under more or less radiation? You seem to have all the data needed to perform a more thorough analysis.

Could it be due to the boundary layer height? Is the boundary layer lower during the monsoon period? Is there any correlation with radiation or temperature? You could perhaps investigate the correlation between delta-GEM and delta-temperature or something like that.

**Response #17**

Thanks for your comments and suggestions. We totally agree with the reviewer that many factors could contribute to the diurnal variations of GEM besides the local emissions. such as wind directions, light radiation, boundary layer height, temperature and some other factors (Li et al.,

2006;Selin, 2009;Horowitz et al., 2017;Travnikov et al., 2017). The Hg(0) reemission from glaciers caused by the high temperature and light radiation might be one of the potential explanation for this change (Faïn et al., 2007;Holmes et al., 2010). We have added more discussions about other possible factors which might affect the diurnal changes of GEM: "**With**

**the increase of ambient temperature and radiation from April to August, the reemission of**

**GEM from glaciers could increase as well. As the snow coverage in the QNNP decreased**

**significantly from the PISM to the ISM period (Figure S4), some of the released Hg may**

**become a source of new GEM from the initial ISM to the final stage of the ISM period. More**

**GEM could be released due to the higher temperature and stronger radiation in the**

**afternoon. However, some other factors such as changes in the PBL heights and in wind**

**directions could also be partly responsible for the diurnal variations of GEM concentrations**

**(Li et al., 2006;Selin, 2009;Horowitz et al., 2017;Travnikov et al., 2017).**" (Line 384-393 in the revised manuscript).

**Comment #18**

Lines 378-388: In Figure 4, could you please use something else than shades of green. It is hard to tell the difference between 1.5 ng/m3.

**Response #18**

Thanks for your suggestion. We have changed the color in Figure 4. Please see the revised manuscript.

**Comment #19**

Lines 410-412: How can you explain that GEM concentrations in air masses originating from the Tibetan Plateau were the highest?

**Response #19**

Thanks for the comment. We think that the high GEM concentrations in air masses originated from the Tibetan Plateau might be caused by some local residential emissions. As we can see from

Figure 5(b), the cluster 2 originated from or passed through the central Tibet, China, where the majority populations in Tibet live in. The local residents usually use the biomass (i.e., yak dung)

for cooking and heating. Previous studies have pointed out that the atmospheric Hg emissions from burning of yak dung could be an important Hg source in Tibet (Rhode et al., 2007;Chen et al.,

2015;Xiao et al., 2015;Huang et al., 2016). We have added this information into the revised manuscript, as follows: "**GEM levels in cluster 2 (23%) were the highest (1.52 ng m$^{-3}$), which**

**originated from or passed through the Tibetan Plateau. The high GEM concentrations could**

**possibly result from the Hg emissions from the burning of yak dung (Rhode et al.,**

**2007;Chen et al., 2015;Xiao et al., 2015;Huang et al., 2016)**" (Line 440-443 in the revised manuscript).

**Comment #20**

Lines 415-417: "The clusters were similar to most of the clusters during the PISM period; however, the GEM concentrations in these clusters were higher than those during the PISM

period". Could you explain why?

**Response #20**

Thank you for the comment. As discussed in Section 3.3.2, the higher GEM concentrations during ISM 2 were likely related with the frequent fire hotspots in the source region. Large amounts of Hg were released from the biomass burning (Finley et al., 2009), leading to the higher

GEM concentration in ISM 2. We have added the following sentence, "**The clusters were similar**

**to most of the clusters during PISM period; however, the GEM concentrations in these**

**clusters were higher than those during the PISM period, which might be caused by the large**

**Hg emissions from frequent fires in the source region during ISM 2 (Finley et al., 2009)**

**(Figure S5).**" (Line 447-450 in the revised manuscript).

**Comment #21**

Lines 452-454: What about Bangladesh? Additionally, you don't really explain why GEM

concentrations increase during the ISM period.

**Response #21**

//Thanks for the comments. We have carefully reviewed the recent publications about atmospheric Hg emission and pollutions in Bangladesh from the Web of Science. However, there are very few literatures about them. Some publications have reported that the air quality in

Bangladesh is very bad (Mondol et al., 2014;Islam et al., 2015;Rana et al., 2016;Ommi et al.,

2017;Rahman et al., 2018). So we think it is possible that the atmospheric Hg emissions in

Bangladesh might also be underestimated similar to Nepal. We have added the following sentences into the revised manuscript: "**Considering the heavy air pollutions in Nepal**

**(Forouzanfar et al., 2015;Rupakheti et al., 2017) and Bangladesh (Mondol et al., 2014;Islam**

**et al., 2015;Rana et al., 2016;Rahman et al., 2018), Nepal and Bangladesh might be the**

**underestimated Hg source regions in the modeling and should be taken into consideration in**

**further study**." (Line 485-489 in the revised manuscript).

//The discussion about the higher GEM in the ISM is provided in Line 437-439, as follows:

"**During the ISM period (Figure 5b-5f), the transport pathways of atmospheric Hg changed**

**signally with the onset of the monsoon and differed strongly from the PISM period.**". We think that frequent fires in the source regions could be an important cause.

**Comment #22**

Line 464: Could you please add the dates for ISM2 here and/or add ISM2 in Figure S3?

**Response #22**

Thanks for the suggestions. The information has been provided in the revised figure.

**Comment #23**

Lines 464-466: Large amounts of PBM "may have been released". In this section and throughout the manuscript, please use the conditional tense to express conjectures/hypotheses.

**Response #23**

Thanks for your suggestion. Revisions have been made accordingly (Line 499-501).

**Comment #24**

Line 471: The discussion is about PBM here, not GOM. Remove reference to GOM.

**Response #24**

We have removed GOM information from the manuscript. Thanks.

**Comment #25**

Line 478: Can you explain this high value? Where did the air masses come from?

**Response #25**

Thanks for your comment and suggestion. We checked the trajectory of the high value, and the trajectory passed through the north of India. This sentence has been revised as follows: "**During**

**the whole monitoring period, the highest GEM concentration reached 3.74 ng m$^{-3}$ (with**

**trajectories passing through the north of India), ~2.5 times higher than the average**

**concentration in the Northern Hemisphere (~1.5-1.7 ng m$^{-3}$ ) (Lindberg et al., 2007;Slemr et**

**al., 2015;Venter et al., 2015).**" (Line 512-515).

**Comment #26**

Lines 484: As mentioned above, 1.3 ng/m$^3$ is at the low end of GEM concentrations reported in the Northern Hemisphere. I agree that there is indeed an influence from South Asia, but concentrations on the QNNP are still fairly low during the PISM. I feel like you should slightly nuance your position.

**Response #26**

We agree that, in general, the GEM concentrations in QNNP are relatively low compared with other monitored values in the background regions of Northern Hemisphere. We have revised the sentence as follows: "**Compared with the ISM period, the GEM concentrations in the PISM**

**period were significantly lower, with a value of 1.31±0.42 ng m$^{-3}$. This value during PISM is**

**not high compared with other background monitoring data in the Northern Hemisphere**."

(Line 518-521).

**Comment #27**

Lines 487-495: Could you possibly add a comparison between PISM and ISM periods in Figure

7? This comparison is the core of your manuscript.

**Response #27**

We have added a comparison between PISM and ISM periods, as follow: "**During the ISM**

**period, the transboundary transport of atmospheric Hg could be strengthened by both**

**monsoon and glacial winds. However, this effect seems to be weaker during the PISM period.**"

(Line 530-532 in the revised manuscript).

**Comment #28**

Line 503: "significant" rather than "considerable".

**Response #28**

We have corrected the word accordingly. Please see Line 538-540 in the revised manuscript.

**Comment #29**

Line 507: Not true everywhere (e.g., Martin et al. 2017).

**Response #29**

We have revised the sentence as follows: "**Atmospheric Hg concentration has been reported**

**to have continuously declined (~1–2% $y^{-1}$) at the monitoring sites in North America and**

**Europe from 1990 to present (Zhang et al., 2016b).** ". Please see Line 542-544 in the revised manuscript.

**Comment #30**

Line 516: Do you know if India, Nepal and Bangladesh have ratified the Minamata Convention on                              Hg?                              Check                              here:

http://mercuryconvention.org/Countries/Parties/tabid/3428/language/enUS/Default.aspx.        Hg emissions are projected to increase in India (Pacyna et al. 2016), what about Nepal and

Bangladesh? You can perhaps strengthen the discussion here.

**Response #30**

Thanks for the information.

//We have reviewed the information in the website carefully. We found that India, Nepal and

Bangladesh have signed the convention, but only India has ratified the convention so far.

//As we replied in Response #21, we have reviewed the recent publications carefully on Web of

Science, but there are very few publications about the Hg emission and Hg concentration in

Bangladesh and Nepal. Some publications have reported that the air quality in Bangladesh is very bad (Mondol et al., 2014;Islam et al., 2015;Rana et al., 2016;Ommi et al., 2017;Rahman et al.,

2018). So we think it is possible that the atmospheric Hg emissions in Bangladesh might also be underestimated, similar to Nepal. We have added this information into the revised manuscript.

Please see Line 562-563.

**Comment #31**

Lines 526-528: Is there a significant difference?

**Response #31**

Yes, in the manuscript, we have performed the statistical analysis to compare the atmospheric

Hg concentrations between PISM and ISM periods, and the results show that there are significant differences between two periods (p<0.001). We have added the statistical information in the revised manuscript. Please see Line 265-271.

**Comment #32**

Lines 544-546: Again, concentrations reported here during PISM are at the low end of concentrations reported in the Northern Hemisphere. Additionally, concentrations are similar to those recently reported at Nam Co station on the Tibetan Plateau (Yin et al. 2018).

**Respond #32**

We agree with the reviewer and we have deleted this sentence from the manuscript.

**Comment #33**

Figure 1: I assume that the red star within the QNNP is the location of the monitoring station.

What about the two other red stars (Lhasa and Xigaze)? Do they represent cities and potential emissions? You should perhaps use a different type of star (monitoring site vs. cities) and make it clear in the caption.

**Response #33**

Yes, Lhasa is the largest city in Tibet, and Xigaze is the second. We have marked these two places in a different symbol to help readers understand the locations of QNNP. Please see the revised Figure 1. Thanks for your suggestions.

**Comment #34**

Figure 2: Could you please add on this Figure the different periods (ISM1-5) you're referring to in Table 1?

**Response #34**

We have highlighted different ISM periods in the revised Figure 2.

**Comment #35**

Figure 3: I can't read the yellow axis, it is too bright. Please use another color. Additionally, what do you mean by GOM or PBM? Is this GOM, PBM, or the sum of the two? It is hard to see the dots and the diurnal cycle for GOM/PBM.

**Response #35**

We have adjusted the color of yellow axis. The hollow and solid dots (in blue) represent the monitored GOM and PBM concentrations, respectively. We have clarified this point in the revised

Figure 3.

**Comment #36**

Figure 4: Which one is GOM, which one is PBM? Add a), b), c) on the Figure and caption.

**Response #36**

We have added the labels accordingly. Thanks.

**Comment #37**

Figure 6: Could you please explain in the caption what these values are? Probability of air passes passing through these regions?

**Response #37**

The figure caption has been revised as: "**Figure 6. Potential source regions and pathways of**

**GEM using the Potential Source Contribution Function (PSCF) method before and during**

**the Indian Summer Monsoon (ISM). PSCF values represent the probability that a grid cell is**

**a source of Hg.**" Please see the revised Figure 6.

**Comment #38**

Table 2: I think you can focus on Asian sites or refer to Figure 1 in Yin et al. (2018). The concentration reported for the Nam Co station is incorrect (Yin et al. 2018).

**Response #38**

In the revised manuscript, we removed the atmospheric Hg monitoring sites out of Asia, which is also suggested by another reviewer. Please see Line 279-280 and the revised Table 2.

**Comment #39**

Figure S4: Could you please add PISM, ISM1-5? Additionally, instead of April-August, is it possible to plot fires during PISM, ISM1-5? It would make it easier to identify whether fires are indeed more frequent in the area of interest during ISM2.

**Response #39**

We have highlighted different ISM periods in Figure S5 and added the fire information as well.

Thanks for the suggestions.

**Reference:**

Burger Chakraborty, L., Qureshi, A., Vadenbo, C., and Hellweg, S.: Anthropogenic mercury flows in

India and impacts of emission controls, Environmental science & technology, 47, 8105-8113, 2013.

Chen, P., Kang, S., Bai, J., Sillanpää, M., and Li, C.: Yak dung combustion aerosols in the Tibetan Plateau: Chemical characteristics and influence on the local atmospheric environment, Atmospheric Research, 156, 58-66,'D'O'I:' 10.1016/j.atmosres.2015.01.001, 2015.

Duan, L., Wang, X., Wang, D., Duan, Y., Cheng, N., and Xiu, G.: Atmospheric mercury speciation in Shanghai, China, Science of the Total Environment, 578, 460-468, 2017.

Faïn, X., Grangeon, S., Bahlmann, E., Fritsche, J., Obrist, D., Dommergue, A., Ferrari, C. P., Cairns, W., Ebinghaus, R., and Barbante, C.: Diurnal production of gaseous mercury in the alpine snowpack before snowmelt, J. Geo. Res. Atoms., 112, 2007.

Finley, B., Swartzendruber, P., and Jaffe, D.: Particulate mercury emissions in regional wildfire plumes observed at the Mount Bachelor Observatory, Atmos. Environ., 43, 6074-6083, 2009.

Forouzanfar, M. H., Alexander, L., Anderson, H. R., Bachman, V. F., Biryukov, S., Brauer, M., Burnett, R., Casey, D., Coates, M. M., and Cohen, A.: Global, regional, and national comparative risk assessment of 79 behavioural, environmental and occupational, and metabolic risks or clusters of risks in 188 countries, 1990–2013: a systematic analysis for the Global Burden of Disease Study 2013, The Lancet, 386, 2287-2323, 2015.

Fu, X., Feng, X., Liang, P., Zhang, H., Ji, J., and Liu, P.: Temporal trend and sources of speciated atmospheric mercury at Waliguan GAW station, Northwestern China, Atmospheric Chemistry and Physics, 12, 1951-1964, 2012.

Giang, A., Stokes, L. C., Streets, D. G., Corbitt, E. S., and Selin, N. E.: Impacts of the minamata convention on mercury emissions and global deposition from coal-fired power generation in Asia, Environmental science & technology, 49, 5326-5335, 2015.

Holmes, C. D., Jacob, D. J., Corbitt, E. S., Mao, J., Yang, X., Talbot, R., and Slemr, F.: Global atmospheric model for mercury including oxidation by bromine atoms, Atmospheric Chemistry and Physics, 10, 12037-12057, 2010.

Horowitz, H. M., Jacob, D. J., Zhang, Y., Dibble, T. S., Slemr, F., Amos, H. M., Schmidt, J. A., Corbitt, E. S., Marais, E. A., and Sunderland, E. M.: A new mechanism for atmospheric mercury redox chemistry: Implications for the global mercury budget, Atmospheric Chemistry and Physics, 17, 6353-6371, 2017.

Huang, J., Kang, S., Guo, J., Zhang, Q., Cong, Z., Sillanpää, M., Zhang, G., Sun, S., and Tripathee, L.: Atmospheric particulate mercury in Lhasa city, Tibetan Plateau, Atmospheric Environment, 142, 433-441, 2016.

Islam, M. F., Majumder, S. S., Al Mamun, A., Khan, M. B., Rahman, M. A., and Salam, A.: Trace metals concentrations at the atmosphere particulate matters in the Southeast Asian Mega City (Dhaka, Bangladesh), Open Journal of Air Pollution, 4, 86, 2015.

Li, M., Dai, Y., Ma, Y., Zhong, L., and Lu, S.: Analysis on structure of atmospheric boundary layer and energy exchange of surface layer over Mount Qomolangma region, Plateau Meteorology, 25, 807-813, 2006.

Lindberg, S., Bullock, R., Ebinghaus, R., Engstrom, D., Feng, X., Fitzgerald, W., Pirrone, N., Prestbo, E., and Seigneur, C.: A synthesis of progress and uncertainties in attributing the sources of mercury in deposition, AMBIO: a Journal of the Human Environment, 36, 19-33, 2007.

Mondol, M., Khaled, M., Chamon, A., and Ullah, S.: Trace metal concentration in atmospheric aerosols in some city areas of Bangladesh, Bangladesh Journal of Scientific and Industrial Research, 49, 263-270, 2014.

Ommi, A., Emami, F., Zíková, N., Hopke, P. K., and Begum, B. A.: Trajectory-based models and remote sensing for biomass burning assessment in Bangladesh, Aerosol Air Qual. Res, 17, 465-475, 2017.

Pacyna, J. M., Travnikov, O., Simone, F. d., Hedgecock, I. M., Sundseth, K., Pacyna, E. G., Steenhuisen, F., Pirrone, N., Munthe, J., and Kindbom, K.: Current and future levels of mercury atmospheric pollution on a global scale, 2016.

Rahman, M. M., Mahamud, S., and Thurston, G. D.: Recent spatial gradients and time trends in Dhaka, Bangladesh air pollution and their human health implications, Journal of the Air & Waste Management Association, 2018.

Rana, M. M., Sulaiman, N., Sivertsen, B., Khan, M. F., and Nasreen, S.: Trends in atmospheric particulate matter in Dhaka, Bangladesh, and the vicinity, Environmental Science and Pollution Research, 23, 17393-17403, 2016.

Rhode, D., Madsen, D. B., Brantingham, P. J., and Dargye, T.: Yaks, yak dung, and prehistoric human habitation of the Tibetan Plateau, Developments in Quaternary Sciences, 9, 205-224, 2007.

Rupakheti, D., Adhikary, B., Praveen, P. S., Rupakheti, M., Kang, S., Mahata, K. S., Naja, M., Zhang, Q., Panday, A. K., and Lawrence, M. G.: Pre-monsoon air quality over Lumbini, a world heritage site along the Himalayan foothills, Atom. Chem. Phys., 17, 11041-11063, 2017.

Selin, N. E.: Global biogeochemical cycling of mercury: a review, Annual Review of Environment and Resources, 34, 43-63, 2009.

Slemr, F., Angot, H., Dommergue, A., Magand, O., Barret, M., Weigelt, A., Ebinghaus, R., Brunke, E.-G., Pfaffhuber, K. A., and Edwards, G.: Comparison of mercury concentrations measured at several sites in the Southern Hemisphere, Atmospheric Chemistry and Physics, 15, 3125-3133, 2015.

Slemr, F., Weigelt, A., Ebinghaus, R., Kock, H. H., Bödewadt, J., Brenninkmeijer, C. A., Rauthe-Schöch, A., Weber, S., Hermann, M., and Becker, J.: Atmospheric mercury measurements onboard the CARIBIC passenger aircraft, Atmospheric Measurement Techniques, 9, 2291-2302, 2016.

Sprovieri, F., Pirrone, N., Bencardino, M., D'Amore, F., Carbone, F., Cinnirella, S., Mannarino, V., Landis, M., Ebinghaus, R., and Weigelt, A.: Atmospheric mercury concentrations observed at ground-based monitoring sites globally distributed in the framework of the GMOS network, Atmospheric Chemistry and Physics, 16, 11915-11935, 2016.

Travnikov, O., Angot, H., Artaxo, P., Bencardino, M., Bieser, J., D'Amore, F., Dastoor, A., Simone, F. D., Diéguez, M. d. C., and Dommergue, A.: Multi-model study of mercury dispersion in the atmosphere: atmospheric processes and model evaluation, Atmospheric Chemistry and Physics, 17, 5271-5295, 2017.

Venter, A., Beukes, J., Van Zyl, P., Brunke, E.-G., Labuschagne, C., Slemr, F., Ebinghaus, R., and Kock, H.: Statistical exploration of gaseous elemental mercury (GEM) measured at Cape Point from 2007 to 2011, Atmospheric Chemistry and Physics, 15, 10271-10280, 2015.

Wan, Q., Feng, X., Lu, J., Zheng, W., Song, X., Han, S., and Xu, H.: Atmospheric mercury in Changbai Mountain area, northeastern China I. The seasonal distribution pattern of total gaseous mercury and its potential sources, Environmental Research, 109, 201-206, 2009.

Wu, Q., Li, G., Wang, S., Liu, K., and Hao, J.: Mitigation options of atmospheric Hg emissions in China, Environmental science & technology, 52, 12368-12375, 2018.

Xiao, Q., Saikawa, E., Yokelson, R. J., Chen, P., Li, C., and Kang, S.: Indoor air pollution from burning yak dung as a household fuel in Tibet, Atmospheric Environment, 102, 406-412,'D'O'I:' 10.1016/j.atmosenv.2014.11.060, 2015.

Zhang, H., Fu, X., Lin, C.-J., Shang, L., Zhang, Y., Feng, X., and Lin, C.: Monsoon-facilitated
characteristics and transport of atmospheric mercury at a high-altitude background site in southwestern
China, Atmospheric Chemistry & Physics, 16, 2016.
Zhang, L., Wang, S., Wang, L., and Hao, J.: Atmospheric mercury concentration and chemical
speciation at a rural site in Beijing, China: implications of mercury emission sources, Atmospheric
Chemistry and Physics, 13, 10505-10516, 2013.